# AUGGEN: GENERATIVE SYNTHETIC AUGMENTATION CAN BOOST FACE RECOGNITION

## ABSTRACT

As machine learning increasingly relies on large amounts of data, concerns about privacy and ethics have grown. Recently, methods for generating synthetic data to augment or replace real datasets have emerged to mitigate these concerns. In this paper, we demonstrate improved performance on a discriminative task when training on a mix of real and synthetic data, compared to training solely on the original real data. Our synthetic data is generated using a novel sampling method based on a conditional generative model and a discriminator, both trained exclusively on the original data, with no need for auxiliary data nor pre-trained foundation models. We consider the challenging task of face recognition, which is well known for its privacy and ethical issues. Using our augmented dataset, we demonstrate consistent improvements over the model trained on the original dataset, on various benchmarks including IJB-C and IJB-B by up to $5\%$ while performing competitively with state-of-the-art synthetic data generation [1].

## 1 INTRODUCTION

As machine learning increasingly relies on data for specific applications, the need for high-quality, accurately labeled datasets is becoming a significant challenge. Moreover, the collection of large datasets required to meet high-performance demands poses even greater challenges when considering privacy and ethical concerns, particularly in sensitive domains such as human face images. One possible solution that recently become popular is synthesizing the data Wood et al. (2021); Azizi et al. (2023); Rahimi et al. (2024); DeAndres-Tame et al. (2024); Bae et al. (2023). The synthetic datasets are generated from different methodologies including 3D-Rendering Graphics and Generative Models (*e.g.*, GANs and Diffusion Models) to name a few. Sometimes because we can generate many examples using our generation methodologies we can even surpass the performance of the model which is trained using real data. An example of this is the work by Wood et al. (2021), which showed that using a 3D-rendering engine and a mesh-based face model enables the generation of precise labels for dense prediction tasks like face landmark localization. This approach can surpass the models trained on the real data as human annotations for such datasets are usually difficult to gather and real datasets are small. The synthetic data can also be created using generative models. Since the introduction of VAEs Kingma (2013), GANs Goodfellow et al. (2020); Karras et al. (2019; 2021), and more recently Diffusion models Song et al. (2020); Karras et al. (2022; 2024); Hoogeboom et al. (2023); Gu et al. (2024), the pace of generative model development has significantly accelerated. These models are often just compared with each other using metrics such as Fréchet Distance (FD) Stein et al. (2023); Heusel et al. (2017), which essentially evaluate how likely generators are to generate samples that look alike to their training datasets, or, in the case of text-to-image generative models, using subjective qualitative metrics like user preferences Esser et al. (2024).

In this paper, we emphasize looking at the usage of the generative models as an *augmentation tool* not *just solely* generating images in respect to metrics such as FD. Currently, the trend is to use generative models Rombach et al. (2022) that are trained on large datasets like LAION-5B Schuhmann et al. (2022). Later, they refine the models for downstream tasks (e.g., classification) using techniques like fine-tuning on separate datasets, prompt engineering, or textual inversion to incorporate augmentation based on generative models Azizi et al. (2023) Trabucco et al. (2024). In the domain of face images, DCFace Kim et al. (2023), authors generated synthetic face images

---

[1]The code and generated datasets will be made available upon publication.

from multiple identities, each exhibiting various intra-class variations. In addition to datasets like CASIA-WebFace for training their method, they employed separate robust face recognition (FR) systems to filter samples based on similarity and other auxiliary networks to balance the generated images across different criteria (e.g., gender, race). It is difficult to determine whether the observed performance gains stem from the massive datasets used to train generative or discriminative models, or from other factors. It is well-known that models exposed to larger and more diverse datasets tend to perform better.

On the contrary, in our approach, as depicted in Figure 1 by using a *single dataset* for training both our discriminative models (*i.e.*, $M = p(\boldsymbol{y}|\mathbf{X})$) and also the generative model (*i.e.*, $G = p(\mathbf{X}|\boldsymbol{y})$ in case of conditional generative model and $G = p(\mathbf{X})$ in case of unconditional version) we want to study the possibility of a performance boost for a discriminative task when it is trained on the mix of the original dataset and the generated one, $M_{mix}$. This performance improvement is in comparison with a discriminative model which was solely trained on the original dataset, $M_{orig}$. This is particularly useful in critical applications in which data is usually scarce.

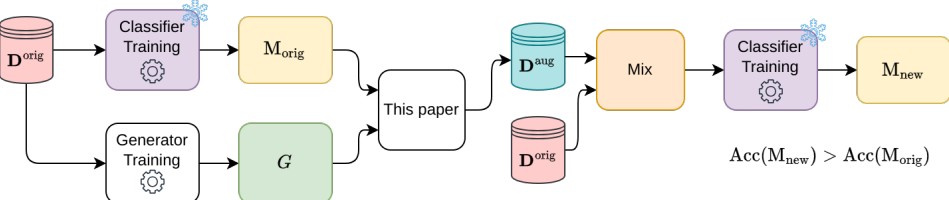

Figure 1: In this paper we explore the use of a generator and discriminator trained on the same dataset to generate useful augmentation of the data that can make the final downstream model more robust (*i.e.*, better performance across diverse benchmarks).

The main contribution of this paper is to validate the following hypothesis for the task of Face Recognition (FR) :

> A generative model, which aims to model $p(\mathbf{X}|\boldsymbol{y})$ can boost the performance of a downstream discriminative model $p(\boldsymbol{y}|\mathbf{X})$ by appropriate informed sampling, and combining the resulting data with the original data that was used for training the generative and discriminative models.

We proposed a novel sampling technique that allows us to validate our hypothesis through extensive FR experiments. To the best of our knowledge, this is the first time that generative image models are considered for augmentation at this scale without the usage of auxiliary models or datasets.

## 2 BACKGROUND

**Usage of Synthetic Data in Computer Vision.** For a smaller number of class variations, (*e.g.*, 2 or 3 classes for classification target), authors in Frid-Adar et al. (2018) train separate generative models. This approach is not scalable for a higher number of classes and variations of our target (*e.g.*, we have thousands of classes for training an FR system). In Azizi et al. (2023), the authors fine-tuned pre-trained diffusion models on ImageNet classes after training on large text-image datasets, demonstrating improved performance on this benchmark through the synthesis of new samples. Recently authors in Kupyn & Rupprecht (2024) introduced the Instance Augmentation method to augment images by redrawing individual objects in the scene retaining their original shape using pre-trained text-to-image models.

**Usage of Synthetic Data in Face Recognition.** Authors in DCFace Kim et al. (2023) have used dual condition latent diffusion models (LDM), one for the style and the other for identity. Similar to our approach, they used CASIA-WebFace Yi et al. (2014) for training their method. By applying auxiliary networks based on race and a separate strong face recognition (FR) system, they filtered the generated images to create their dataset. In Sevastopolskiy et al. (2023), the authors collected a large set of unlabeled face images from various ethnicities and pre-trained a StyleGAN2-ADA (SG2) model Karras et al. (2020). They then trained an encoder to map images to SG2's latent space. By transferring the encoder's weights to the face recognition (FR) network, they demonstrated a

bias-mitigated version of the final FR system. GANDiffFace Melzi et al. (2023) uses a pre-trained Stable Diffusion model Rombach et al. (2022) trained on the large LAION-5B dataset Schuhmann et al. (2022). The method involves two steps: first, synthesizing identities using StyleGAN3 Karras et al. (2021) and transforming them in its latent space. Then, by applying Stable Diffusion and DreamBooth Ruiz et al. (2023) fine-tuning, they introduce more intra-class variability. In IDiff-face Boutros et al. (2023), a LDM was conditioned on the embedding space of a face recognition (FR) system trained on the MS1Mv2 dataset to generate their data. Authors in DigiFace1M Bae et al. (2023) used 3D rendering pipelines to generate various intra-class variabilities of different identities by accessorizing them and rendering the final 3D model in different poses, expressions, and lighting conditions. Recently in Rahimi et al. (2024), authors used off-the-shelf image-to-image translation methodologies without any identity information and demonstrated performance improvement in comparison to the original DigiFace1M.

As discussed here, while most methods rely on auxiliary datasets and models to show improvement in some datasets, we take a different approach, demonstrating a performance boost without using any additional models or data in most of the FR benchmarks.

## 3 METHODOLOGY

Figure 2 provides an overview of our proposed methodology. Our approach involves using the features from a discriminator $M_{orig}$ and the generated images from a generator $G$, both trained on a single dataset. We condition $G$ so that the generated images can be treated as new classes, effectively augmenting the original dataset. First, we outline a general problem formulation for both the discriminator (Classifier Training and its output) and the generator (Generator Training block and its output) in subsection 3.1 and subsection 3.2 respectively. We then present our main contribution: generating new classes (Finding Weights in the figure) to augment real datasets with the generated images in the latter.

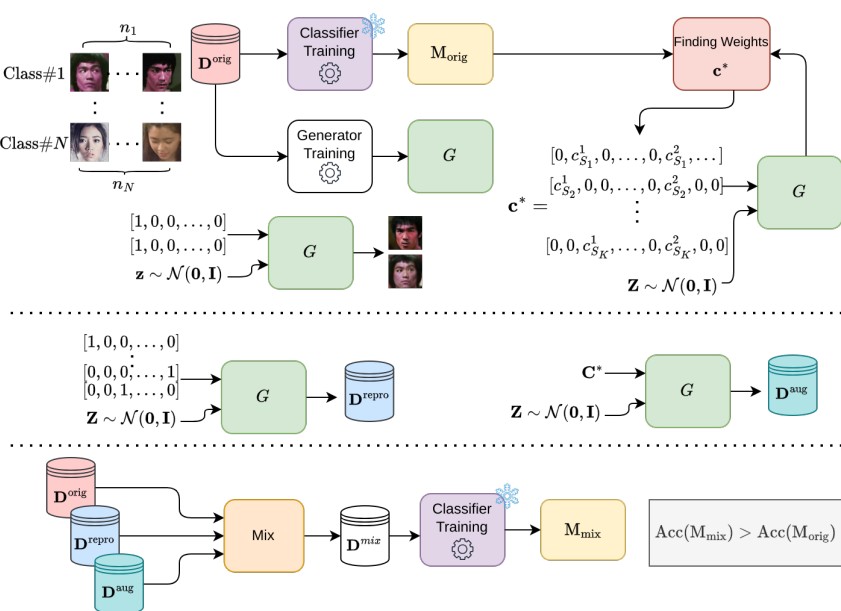

Figure 2: Overview diagram of AugGen, we used a single labeled dataset, $D^{orig}$, for training a class-conditional generator, $G(\mathbf{Z}, c)$ and also a discriminative model, $M_{orig}$. Later with both of these models, we find new condition vectors, $C^*$ that will lead to a new dataset, $D^{aug}$ (when we are giving the new condition to the generator and sampling from it). The conditions are set in a way that the newly generated dataset would be beneficiary for boosting the performance of the $M_{orig}$ when we augment it with the original dataset. This is done without relying on any auxiliary dataset/model.

## 3.1 Discriminative Model

Assume that we have a dataset, $\mathbf{D}_{\text{orig}}$, consist of k pairs of image and label, $\{(\mathbf{X}_0, y_0), (\mathbf{X}_1, y_1), \ldots, (\mathbf{X}_k, y_k)\}$. $\mathbf{X}_i$ is an image of the form $\mathbb{R}^{H \times W \times 3}$, for simplicity here we assume that the H and W are fixed for all of the $k$ sample pairs in our dataset. $y_i$ is a scalar that depicts the label of images from a fixed set of $l$ possible values (*e.g.*, $\{0, 1, ..., l\}$), $\mathbb{L}$, which $l \ll$ k. Here the goal of a discriminative model is to learn the conditional probability $p(\boldsymbol{y}|\mathbf{X})$ using the samples in $\mathbf{D}^{\text{orig}}$. For example here the $\mathbf{D}^{\text{orig}}$ can be the ImageNet Russakovsky et al. (2015) or CASIA-WebFace Yi et al. (2014) dataset. The discriminative model's task is to identify the most likely class of the image. In most cases this enforces the features (*i.e.*, usually output of penultimate layer, $\boldsymbol{e}$) for the similar images to be closer together according to a measure, $m$ (*e.g.*, Cosine Similarity, the lower the output of $m(\boldsymbol{e}_1, \boldsymbol{e}_2)$ the more similar the features $\boldsymbol{e}_1$ and $\boldsymbol{e}_2$ are). This is usually learned by a mapping function $f_{\theta_{\text{dis}}} : \mathbf{X} \to \boldsymbol{y}$ parameterized by $\theta_{\text{dis}}$ which uniquely describes the architecture and the parameters of the $f$. The $\theta_{\text{dis}}$ is learned through empirical risk minimization as follows:

$$\theta_{\text{dis}}^* = \underset{\theta_{\text{dis}} \in \Theta_{\text{dis}}}{argmin} \ \mathbb{E}_{(\mathbf{X},y) \sim \mathbf{D}^{\text{orig}}}[\mathcal{L}_{\text{dis}}(f_{\theta_{\text{dis}}}(\mathbf{X}), \boldsymbol{y})] \tag{1}$$

Where for a classification problem the, $\mathcal{L}_{\text{dis}}$, is usually in the form of Cross-Entropy and the Expectation here is being calculated by drawing sample pairs from, $\mathbf{D}^{\text{orig}}$. Here we refer to possible hyperparameters for calculating the $\theta_{\text{dis}}^*$ as $h_{\text{dis}}$, which tries to abstract out the processes such as Learning rates and its schedules, number of epochs for training and other complexities in the real world training of a neural network. As depicted in Figure 2, The outcome of this process is a model, $M_{\text{orig}}$, (*i.e.*, $f_{\theta_{\text{dis}*}}$), which tries to reflect that the model is trained on the $\mathbf{D}_{\text{orig}}$.

## 3.2 Generative Model

Generative models aim to capture the underlying distribution of the dataset given some samples, such that new samples can be drawn from it. Here we present general problem formulation in the context of the diffusion models, in which we also demonstrate the experiments using these types of models in section 4. The idea of the diffusion models Song et al. (2020); Anderson (1982) is that by sequentially adding noise to the data (*i.e.*, $\mathbf{X}_i$s ) and learning a denoiser/score function, S. This is done to gradually learn to go from a complete white Gaussian noise (*i.e.*, in the sampling process) to the data distribution, by adding noise to the data in different scales and learning the noise that was added to the data during the training process (*i.e.*, supervision). Following Karras et al. (2024; 2022) formulation, S, the denoiser function can be learned in two stages. The first stage is that given a noise level, $\sigma$, we optimize the parameters of the denoiser (*i.e.*, $\theta_{\text{den}}$) by adding a noise $\mathbf{N}^{h \times w \times c}$ which is dependent on $\sigma$ and removing it by the denoiser function as follows:

$$\mathcal{L}(S_{\theta_{den}}; \sigma) = \mathbb{E}_{(\mathbf{X},y) \sim \mathbf{D}^{\text{orig}}, \mathbf{N} \sim \mathcal{N}(\mathbf{0}, \sigma\mathbf{l})}[||S_{\theta_{den}}(E_{\text{VAE}}(\mathbf{X}) + \mathbf{N}; \text{c}(y), \sigma) - \mathbf{X}||_2^2] \tag{2}$$

Here the image and its corresponding label, $(\mathbf{X}, y)$ are sampled from, $\mathbf{D}^{\text{orig}}$. The $\mathbf{X}$ is passed to a VAE encoder, $\mathbf{Z}^{h \times w \times c} = E_{\text{VAE}}(\mathbf{X})$ as we are working on the latent diffusion paradigm Rombach et al. (2022). Later this $\mathbf{Z}$ can be transformed back to image space using the VAE's decoder, $\mathbf{X} \simeq D_{\text{VAE}}(E_{\text{VAE}}(\mathbf{X}))$. Latent diffusion models are mostly popular because of lower computational cost with respect to pixel-level diffusion models, especially in higher resolution. The denoising is done in the latent space of the VAE and later decoded to the pixel space. The denoiser function takes the noisy input, noise scale, and the class condition, $c(y)$ as input and tries to estimate the original latent, $E_{\text{VAE}}(\mathbf{X})$. The second stage is to iterate over different data-dependent noise scales which reduces the final optimization target to:

$$\theta_{den}^* = \underset{\theta_{den} \in \Theta_{den}}{argmin} \ \mathbb{E}_{\sigma \sim \mathcal{N}(\mu, \sigma^2)}[\lambda_\sigma \mathcal{L}(S_{\theta_{den}}; \sigma)] \tag{3}$$

Which $\lambda_\sigma$ is noise scale dependent weight, and $\mu$ and $\sigma$ in $\mathcal{N}(\mu, \sigma^2)$ are empirically set to focus the training on more important noise levels for the latent space of a VAE Karras et al. (2024; 2022); Rombach et al. (2022).

In our formulation, we used one-hot encoding for the $c$ function, meaning that, if the dataset $\mathbf{D}^{\text{orig}}$ contains $l$ unique labels, $c$ maps each $i \in \{0, 1, \ldots, l\}$ to a unique vector of size $l$ that the value corresponding to the label is 1, while the rest are set to 0. After training the conditional denoiser (*i.e.*, $S_{\theta_{\text{den}}}$), using Equation 3, as depicted in the middle of Figure 2, we can sample from the generator in two ways.

1. During sampling, we pass the condition vector $c$ as it was during the training of S, i.e., one-hot vectors representing the classes.

2. The condition $c^*$ is different from the values used during training.

As an example, depicted in Figure 2, when we pass $c$, one-hot condition vector corresponding to the first class, we expect the generator to synthesize samples that are highly similar to the first class in the, $D^{\text{orig}}$. We refer to this dataset that tries to reproduce the, $D^{\text{orig}}$, the $D^{\text{repro}}$. In this paper, we explored the ways of conditioning a generator with the values that it has not seen before, and how we can use the model $M_{\text{orig}}$ that was trained using the dataset $D^{\text{orig}}$ to generate samples that can be used to make $M_{\text{mix}}$ more robust (*i.e.*, consistently more performing in various benchmarks). Our goal is to synthesize a new class by combining two conditions that the generator has seen before (*i.e.*, the one-hot condition during training). To this end, given the one-hot condition of two classes, $i$ and $j$, namely, $c^i$ and $c^j$, we seek to find $\alpha$ and $\beta$ such that the condition vector of a hypothetically new class would be:

$$c^* = \alpha c^i + \beta c^j \tag{4}$$

For ease of notation here we denote the generation process of the trained denoiser by a function $G$, $\mathbf{X}^i = G(\mathbf{Z}, c^i)$, which involves giving as input the $\mathbf{Z} \sim \mathcal{N}(\mathbf{0}, \mathbf{I})$ and condition vector $c$ to generator to denoise the noisy latent iteratively based on a noise scheduler and finally decode it using $D_{\text{VAE}}$ back to image space. For finding the $\alpha$ and $\beta$ that produces a class that is dissimilar to the source classes that we are mixing (*i.e.*, $i$ and $j$ in Equation 4) and also similar within each other (*i.e.*, when we give the same condition to $G$ we expect the model to generate images of the same class) we formulate the problem as a search grid.

We set the $\alpha$ and $\beta$ to some possible combinations for linear space of the values between $0.1$ to $1.1$. For example, possible combinations would be $\alpha = 0.3, \beta = 0.5$ or $\alpha = 1.1, \beta = 0.4$. We denote $\mathbb{W}$, the set which contains possible values of $\alpha$ and $\beta$. We also select some subset of $\mathbb{L}$ and call it $\mathbb{L}_s$, for the set to contain some specific classes. Then we randomly select two values from the $\mathbb{L}_s$ namely $i$ and $j$. Later for each $(\alpha, \beta) \in \mathbb{W}$ we calculate the Equation 4, to get the $c^*$. For $K$ times we generate three types of images. The first two is the reproduction dataset, $D^{\text{repro}}$ as before by setting the conditions to $c^i$ and $c^j$, to get $\mathbf{X}^i = G(\mathbf{Z}, c^i)$ and $\mathbf{X}^j = G(\mathbf{Z}, c^j)$. Finally the third one is $\mathbf{X}^* = G(\mathbf{Z}, c^*)$. By passing the generated images to the $f_{\theta_{\text{dis}*}}$ (*i.e.*, our discriminator which was trained on the $D^{\text{orig}}$) we get the features, $e^i$, $e^j$ and $e^*$. As mentioned previously we seek to maximize the dissimilarity between generated images so that we can treat the new sample $\mathbf{X}^*$ as a new class. For this, we use a dissimilarity measure, $m_d$ which the higher its absolute value it produces the more dissimilar the inputs are. Later we calculate this measure for each of the reproduced images of the existing classes in respect to the new class, $d_i = m_{\text{d}}(e^i, e^*)$ and $d_j = m_{\text{d}}(e^j, e^*)$, here we define the total dissimilarity between the reproduced classes and the newly generated class as $m_{\text{d}}^{\text{total}} = |d_i| + |d_j|$. As mentioned earlier we repeat this process $K$ times, this means that we get $K$ different $\mathbf{X}^*$. We also want that $\mathbf{X}^*$ to be as similar as possible to each other so we can assign the same label/class to them for a fix $\alpha$ and $\beta$. To this end, we also calculate a similarity measure, $m_s$, in which the higher the absolute output of this measure is the more similar their input is. We calculate it between the $K$ generated $\mathbf{X}^*$ as $m_{\text{s}}^{\text{total}}$. We hypothesize and verify later with our experiments that the good candidates for $\alpha$ and $\beta$ are the ones that have a high value of the $m^{\text{total}} = m_{\text{s}}^{\text{total}} + m_{\text{d}}^{\text{total}}$. This search for $\alpha$ and $\beta$ is presented in the Algorithm 1.

After finding candidate values for $\alpha$ and $\beta$, by randomly selecting classes from $\mathbb{L}$, and calculating $c^*$, we can generate images that represent a hypothetically new class. The output of this process is what we call generated augmentations of the $D^{\text{orig}}$, or $D^{\text{aug}}$ as depicted in the middle row of the Figure 2. Later as depicted in the last line of Figure 2, in the experiments, we demonstrated that combining this generated dataset with the $D^{\text{orig}}$ can make the downstream discriminative model more robust. Additionally, in Appendix C , we experimentally demonstrate that common metrics for evaluating generator performance do not correlate with the final performance on downstream task.

## 4 EXPERIMENTS

In this section, we demonstrate the effectiveness of our proposed method for generating augmentations. We consider the problem of Face Recognition (FR). As previously mentioned, the challenges associated with the large datasets required for training modern FR systems are significant. Therefore, achieving better performance with smaller datasets is advantageous. Here we show that in

---

**Algorithm 1** Grid search for $\alpha$ and $\beta$

---

**Require:** Search range for $\alpha, \beta \in [0.1, 1.1], \mathbb{L}_s \subseteq \mathbb{L}$, $K$: Number of iterations.
**Require:** $G(.,.)$: Class-conditional Generator trained on $\mathrm{D}^{\mathrm{orig}}$
**Require:** $f_{\theta_{\mathrm{dis}}^*}$: Discriminator trianed on $\mathrm{D}^{\mathrm{orig}}$
      **Output:** $\alpha^*$ and $\beta^*$
1: Create set $\mathbb{W} = \{(\alpha, \beta) \mid \alpha, \beta \in [0.1, 1.1]\}$
2: Randomly select two values $i$ and $j$ from $\mathbb{L}_s$
3: Create empty set $\mathbb{M}$.
4: **for** each $(\alpha, \beta) \in \mathbb{W}$ **do**
5:     $\boldsymbol{c}^* = \alpha \boldsymbol{c}^{\mathrm{i}} + \beta \boldsymbol{c}^{\mathrm{j}}$
6:     Create empty set $\mathbb{F}$.
7:     **for** k = $1, \ldots, K$ **do**
8:         Get Images : $\mathbf{X}^i = G(\mathbf{Z}, \boldsymbol{c}^i), \mathbf{X}^j = G(\mathbf{Z}, \boldsymbol{c}^j), \mathbf{X}^* = G(\mathbf{Z}, \boldsymbol{c}^*)$
9:         Get Features: $\boldsymbol{e}^i, \boldsymbol{e}^j, \boldsymbol{e}^* = f_{\theta_{\mathrm{dis}*}}(\mathbf{X}^i), f_{\theta_{\mathrm{dis}*}}(\mathbf{X}^j), f_{\theta_{\mathrm{dis}*}}(\mathbf{X}^*)$
10:        Add $\boldsymbol{e}^*$ to $\mathbb{F}$
11:        Dissimilarity measures: $d_i = m_{\mathrm{d}}(\boldsymbol{e}^i, \boldsymbol{e}^*), \ d_j = m_{\mathrm{d}}(\boldsymbol{e}^j, \boldsymbol{e}^*)$
12:        Total dissimilarity: $m_{\mathrm{d}}^{\mathrm{total}} = |d_i| + |d_j|$
13:     **end for**
14:     $m_{\mathrm{s}}^{\mathrm{total}} = 0$
15:     $\forall p, q \in \mathbb{F} | p \neq q$ Calculate $m_{\mathrm{s}}(\boldsymbol{e}^p, \boldsymbol{e}^q)$ and add it to $m_{\mathrm{s}}^{\mathrm{total}}$
16:     Final measure: $m^{\mathrm{total}} = m_{\mathrm{s}}^{\mathrm{total}} + m_{\mathrm{d}}^{\mathrm{total}}$ and add it to $\mathbb{M}$.
17: **end for**
18: Return $\alpha^*$ and $\beta^*$ that the $m^{\mathrm{total}}$, in $\mathbb{M}$ is high.

---

various benchmarks training with our synthetically generated augmentation is beneficiary for the downstream model with respect to a model trained solely on the real dataset, $\mathrm{D}^{\mathrm{orig}}$.

### 4.1 EXPERIMENT SETUP

Here we set the dataset $\mathrm{D}^{\mathrm{orig}}$ to CASIA-WebFace Yi et al. (2014). This dataset contains $10,572$ identities and also for each identity some variations (*e.g.*, same identity in different lighting, expression, and poses).

**Discriminative Model** For training the discriminator and a fair comparison between different methods, we trained an FR system consisting of a ResNet50 backbone as modified in ArcFace's implementation Deng et al. (2019), with the AdaFace Kim et al. (2022) head for margin loss. We trained a separate network multiple times under the same conditions, like the same number of GPUs for training them (unless otherwise mentioned), the same learning rate schedule (*i.e.*, same $h_{\mathrm{dis}}$ in subsection 3.1 refer to the appendix Table 7 for the details of the $h_{\mathrm{dis}}$). The only variable in the multiple training iterations is the seed which controls the initialization of the weights of the network and other sources of randomness like the order in which the empirical minimization algorithm is observing the training data which leads to slightly different results. The number of iterations was set between 2 and 4, based on the observed performance variance of the final downstream model. These hyper-parameters have been used to train a FR backbone for each synthetic dataset such as, the original DigiFace1M dataset (from 3D graphics) and its translated RealDigiFace versions Rahimi et al. (2024) (*i.e.*, Hybrid, 3D graphics and post-processing), and the two Diffusion-based DCFace Kim et al. (2023) and IDiff-Face Boutros et al. (2023) datasets. We also applied common augmentations for the FR task, such as photometric, cropping, and low-resolution augmentations.

**Generative Model** To train our generative model, we used a variant of the latent diffusion formulation Karras et al. (2022; 2024). In this case, the one-hot condition vectors $\boldsymbol{c}^{10572}$ have a size of 10,572, corresponding to the number of classes in $\mathrm{D}^{\mathrm{orig}}$. We train two versions of the latent diffusion model (LDM) from scratch, labeled small and medium, to analyze the impact of network size and training iterations on the final performance, following the approach outlined in the original papers Karras et al. (2024; 2022). See Appendix B for additional details on training the generator.

**Grid Search** As presented in the Algorithm 1 we need to find an appropriate $\alpha$ and $\beta$ for generating useful augmentations based on the generator trained in the previous section. For this we set the $\mathbb{L}_s$ to the classes from the generator which are presented more than the median number of

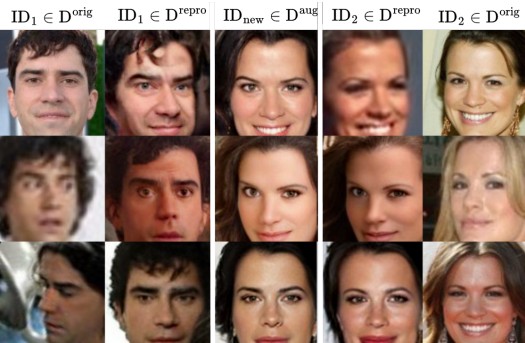

$ID_1 \in D^{orig}$   $ID_1 \in D^{repro}$   $ID_{new} \in D^{aug}$   $ID_2 \in D^{repro}$   $ID_2 \in D^{orig}$

Figure 3: Random sample, from left to right, the first column is variations of a random ID, 1, in the, $D^{orig}$, the second column is the reproduction of the same ID in the first column using the generator, when we put the conditions to corresponding one-hot $G(\mathbf{Z}, \boldsymbol{c}_1)$, The last two columns are the same but for different ID and the middle column representing the new class/identity by generating image using $G(\mathbf{Z}, \boldsymbol{c}^*)$.

samples per class, we empirically observed that these classes are better reproduced when we were generating $D^{repro}$. Later we set the $\mathbb{W}$ to $\{0.1, 0.2, \dots, 1.0, 1.1\}$ for searching $\alpha$ and $\beta$ to calculate the new condition vector $\boldsymbol{c}^*$. Closely related on how the FR models are being trained, especially the usage of the margin loss, (*i.e.*, AdaFace Kim et al. (2022) or ArcFace Deng et al. (2019)), we set the measure for dissimilarity between the features of the two sample images, $\mathbf{X}^1$ and $\mathbf{X}^2$, to cosine similarity which calculating, $m_d = \frac{\boldsymbol{e}^1 . \boldsymbol{e}^2}{||\boldsymbol{e}^1||||\boldsymbol{e}^2||}$. Note that the $\boldsymbol{e}$s were calculated using a discriminator that was trained solely on the $D^{orig}$. We treat the values of the measure in such a way that the higher the output of the measure the more it is reflecting its functionality (*i.e.*, the larger the measure for dissimilarity is the more dissimilar the inputs are). Accordingly, we set the similarity measure to $m_s = 1 - |\frac{\boldsymbol{e}^1 . \boldsymbol{e}^2}{||\boldsymbol{e}^1||||\boldsymbol{e}^2||}|$, which again reflects that the inputs are more similar if the output of this measure is closer to 1. We iterate multiple choices of the, $i$ and $j$ and average our $m^{total}$ for each of the choices. A sample of the output of this process is depicted in Figure 4. Here we observe that by increasing the $\alpha$ and $\beta$ from $(0.1, 0.1)$ to between $(0.7, 0.7)$ and $(0.8, 0.8)$ the measure increases and after that, it will decrease when we go toward $(1.1, 1.1)$, we specifically interested in the $\alpha = \beta$ line as we do not want to include any bias regarding the classes that we randomly choose. We consider three sets of values for $(\alpha, \beta)$, $(0.5, 0.5)$, $(0.7, 0.7)$ and $(1.0, 1.0)$ corresponding to the $m^{total}$ of $1.48$, $1.58$ and $1.53$ respectively. We set the output of Algorithm 1 to $(0.7, 0.7)$. We will show quantitatively the effectiveness of this measure in the final performance of discriminator when we train it on the synthetically generated dataset using various $\alpha$ and $\beta$ in subsection 4.3.

**Synthetic Dataset**  For generating the reproduction dataset $D^{repro}$, we set the condition for each of the $10,572$ classes in the original CASIA-WebFace dataset to the generator. The number of samples per class is $50$ unless mentioned otherwise. For generating $D^{aug}$ we randomly sampled $10,000$ combination of the $\mathbb{L}_s$, $\binom{Card(\mathbb{L}_s)}{2}$, (samples with more than the median number of sample/class in the original CASIA-WebFace), and fixed them for all the experiments. Later by setting the $\alpha$ and $\beta$ to candidate values found in the previous section, $(0.7, 0.7)$, we generated $50$ sample per $10,000$ selected classes. In Figure 3, some samples of the generated images are shown. Here the first and last columns are the examples of the two classes in the original dataset. The second and 4-th columns are the reproduction of the identities in the first and last column respectively (*i.e.*, $D^{repro}$). Each line is generated using the same seed (source of randomness in the generator), and finally, the middle column (3rd) is the $D^{aug}$ which is generated by $\mathbf{X}^* = G(\mathbf{Z}, \boldsymbol{c}^*)$ when we calculate the $\boldsymbol{c}^*$. We can observe that the middle column identity is slightly different from the source classes while being coherent when we generate multiple examples of this new identity. This might be one of the reasons why our augmentation is improving the final performance. Please refer to the appendix Appendix D for more samples.

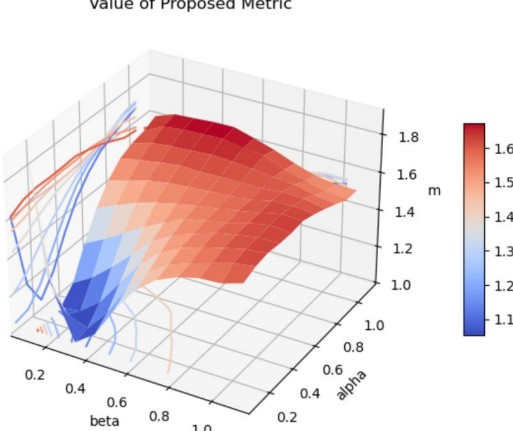

Figure 4: The value of the proposed measure $m^{\text{total}}$ for setting the candidate values of $\alpha$ (x axis) and $\beta$ (y axis). Here for each $\alpha$ and $\beta$ and our 100 combination of $\mathbb{L}_s$ we calculated the $m^{\text{total}}$ by setting the $K$ in Algorithm 1 to 10.

## 4.2 FACE RECOGNITION BENCHMARKS

We show that our synthetic augmentation is boosting the performance of a model trained on the real dataset in all the benchmarks. For this purpose, we evaluated against two sets of FR benchmarks. The first set consists of LFW Huang et al. (2008), CFPFP Sengupta et al. (2016), CPLFW Zheng & Deng (2018), CALFW Zheng et al. (2017), AgeDB Moschoglou et al. (2017), which includes mainly high-quality images with various lighting, poses, and ages Table 1. The second set involves benchmarks consisting of medium to low-quality images from a realistic and challenging FR scenario (NIST IJB-B/C) Maze et al. (2018); Whitelam et al. (2017) Table 2. In table Table 2, we mainly show the verification accuracy for two thresholds which are usually used in real-world scenarios when the FR systems are being deployed, namely TPR@FPR=1-e-06 and TPR@FPR=1e-05 for both IJB-B and IJB-C. Please refer to appendix Appendix A for the IJB-B and IJB-C results for all usual FPRs. In the Table 1 and Table 2, the **Aux** column depicts that if the method under study used any auxiliary model for the generation of the dataset other than the $D^{\text{orig}}$. The ideal value for this column is N which refers to not using any auxiliary model/datasets. The $n^s$ and $n^r$ depict the number of synthetic and real images used for training the discriminative model. The final values for the benchmarks are reported as mean and std of the observed numbers when we are changing only the seed as discussed before for stronger conclusions. Table 2 and Table 1 consist of two parts separated by a double horizontal line. The upper part refers to fully synthetic FR training, without including any real data. The second part consists of fully real FR training, as well as mixed training which consists of the same real dataset and the synthetic data from three methods: the proposed AugGen, DCFace Kim et al. (2023) and IDiffface Boutros et al. (2023). This ensures a fair comparison, as all the mentioned methods are generative model-based, unlike DigiFace1M and RealDigiFace1M, which are used for fully synthetic training. Here for *each* part of the table **bold** and underline text are presenting best and second best respectively. In the second part, if augmentation with the real CASIA-WebFace performed better than solely training with the CASIA-WebFace (middle part of both tables) the cell is shaded in **gray**. For the less challenging benchmarks in Table 1, we observe that although our method consists of a smaller number of samples and does not use any auxiliary model/data we are performing competitively with other state-of-the-art (SOTA) methods/datasets. In the second part of this table we are observing mainly all methods we combined with the CASIA-WebFace are boosting the discriminator which is solely trained on the CASIA-WebFace. In the Table 2, we demonstrate better general performance being the best or second to best in most benchmarks although our dataset were generated for augmentation by design. By observing the results after the augmentation (second part of the table) we are the only method that consistently performs better than the baseline. One interesting finding was the performance drop of the model when it was combined with the CASIA-WebFace in the Table 2. But we are observing that *consistently in all of the benchmarks, our augmentation methodology is boosting the baseline*. We demonstrate

that although we did not use any auxiliary model/data our synthetic dataset performed competitively with other state-of-the-art methods or even outperformed them in some cases.

Table 1: Comparison of the fully synthetic FR training (upper part), fully real FR training (middle), and mixed FR training (bottom) using CASIA-WebFace, when the models are evaluated in terms of accuracy against standard FR benchmarks, namely LFW, CFPFP, CPLFW, AgeDB and CALFW with their corresponding protocols. Here $n^s$ and $n^r$ depict the number of Synthetic and Real Images respectively and Aux depicts whether the method for generating the dataset uses an auxiliary information network for generating their datasets (Y) or not (N).

| Method/Data | Aux | $n^s$ | $n^r$ | LFW | CFP-FP | CPLFW | AgeDB | CALFW | Avg |
|---|---|---|---|---|---|---|---|---|---|
| DigiFace1M | N/A | 1.22M | 0 | 92.43±0.00 | 74.64±0.06 | 82.57±0.43 | 75.72±0.51 | 69.48±1.32 | 78.97±0.44 |
| RealDigiFace | Y | 1.20M | 0 | 93.88±0.19 | 76.95±0.17 | 85.47±0.06 | 77.57±0.07 | 72.82±0.59 | 81.34±0.02 |
| IDiff-face | Y | 1.2M | 0 | 97.45±0.05 | 77.07±0.34 | 80.48±0.63 | 87.26±0.05 | 81.15±0.61 | 84.68±0.05 |
| DCFace | Y | 0.5M | 0 | 98.33±0.07 | 82.50±0.11 | 90.28±0.20 | 91.52±0.05 | 89.67±0.36 | 90.46±0.07 |
| DCFace | Y | 1.2M | 0 | **98.79±0.11** | 84.20±0.34 | **91.19±0.01** | **92.50±0.07** | **91.22±0.06** | **91.58±0.09** |
| AugGen, $D^{aug}$ (Ours) | N | ∼0.6M | 0 | 97.69±0.03 | 81.55±0.03 | 86.88±0.46 | 88.49±0.04 | 83.74±0.01 | 87.67±0.09 |
| AugGen $D^{repro}$ (Ours) | N | ∼0.6M | 0 | 98.60±0.02 | **85.26±0.14** | 91.13±0.14 | 90.54±0.16 | 87.69±0.19 | 90.64±0.07 |
| CASIA-WebFace | N/A | 0 | ∼0.5M | 99.21±0.18 | 87.85±1.72 | 95.69±1.16 | 92.78±0.47 | 92.71±0.96 | 93.65±0.89 |
| IDiff-face | Y | 1.2M | ∼0.5M | **99.53±0.07** | **89.92±0.01** | **96.91±0.27** | 93.64±0.16 | **94.28±0.04** | **94.86±0.02** |
| DCFace | Y | 0.5M | ∼0.5M | 99.43±0.08 | 89.44±0.42 | 96.67±0.16 | 93.82±0.04 | 94.24±0.15 | 94.72±0.09 |
| AugGen $D^{aug}$ (Ours) | N | ∼0.2M | ∼0.5M | 99.41±0.08 | 89.32±0.02 | 96.41±0.09 | 93.13±0.03 | 93.63±0.15 | 94.38±0.00 |

Table 2: Comparison of the fully synthetic FR training (upper part), fully real FR training (middle), and mixed FR training (bottom) using CASIA-WebFace, when the models are evaluated against challenging FR benchmarks with their standard protocols: on IJB-B (B) and IJB-C (C) in terms of True Positive Rate (TPR) using two thresholds set for two practical False Positive Rates (FPRs), and also on TinyFace in terms of Rank-1 accuracy (TR1). Here $n^s$ and $n^r$ depict the number of Synthetic and Real Images respectively and Aux depicts whether the method for generating the dataset using an auxiliary information network for generating their datasets

| Method/Data | Aux | $n^s$ | $n^r$ | B-1e-6 | B-1e-5 | C-1e-6 | C-1e-5 | TR1 |
|---|---|---|---|---|---|---|---|---|
| DigiFace1M | N/A | 1.22M | 0 | 15.31±0.42 | 29.59±0.82 | 26.06±0.77 | 36.34±0.89 | 32.30±0.21 |
| RealDigiFace | Y | 1.20M | 0 | 21.37±0.59 | 39.14±0.40 | 36.18±0.19 | 45.55±0.55 | 42.64±1.70 |
| IDiff-face | Y | 1.2M | 0 | 26.84±2.03 | 50.08±0.48 | 41.75±1.04 | 51.93±0.89 | 45.98±0.61 |
| DCFace | Y | 0.5M | 0 | 29.74±2.25 | **57.55±0.76** | **51.64±1.55** | **64.58±1.01** | 42.85±0.07 |
| DCFace | Y | 1.2M | 0 | 22.48±4.35 | 47.84±6.10 | 35.27±10.78 | 58.22±7.50 | 45.94±0.01 |
| Auggen $D^{aug}$ (Ours) | N | ∼0.6M | 0 | **32.67±1.17** | 51.52±0.69 | 47.74±0.47 | 58.07±0.48 | 48.10±0.05 |
| Auggen $D^{repro}$ (Ours) | N | ∼0.6M | 0 | 15.71±3.12 | 45.97±4.64 | 31.54±6.65 | 58.61±3.89 | 53.61±0.47 |
| CASIA-WebFace | N/A | 0 | ∼0.5M | 1.16±0.08 | 5.61±1.64 | 0.83±0.10 | 5.86±1.31 | 58.01±0.28 |
| IDiff-face | Y | 1.2M | ∼0.5M | 0.89±0.07 | 5.80±0.63 | 0.70±0.11 | 7.46±2.08 | **59.32±0.34** |
| DCFace | Y | 0.5M | ∼0.5M | 0.26±0.11 | 1.59±0.51 | 0.18±0.07 | 1.54±0.59 | 56.60±0.41 |
| Auggen $D^{aug}$ (Ours) | N | ∼0.2M | ∼0.5M | **1.29±0.01** | **8.21±1.38** | **1.43±0.22** | **9.67±1.01** | 58.01±0.50 |

## 4.3 Effectiveness of Grid Search

We study the effectiveness of our proposed method in Algorithm 1 which tries to find the suitable condition weights, $\alpha$, and $\beta$. We compare with four sets of values:

- Rand: $\alpha$ and $\beta$ were selected randomly for $10,000$ mixture of identities from the set of $\{0.1, 0.3, 0.5, 0.7, 0.9, 1.0, 1.1\}$.

- Half: $\alpha$ and $\beta$ set to $0.5$ for all $10,000$ random mixture of identities selected from $\mathbb{L}_s$.

- Full: $\alpha$ and $\beta$ set to $1$ for all $10,000$ random mixture of identities selected from $\mathbb{L}_s$.

- Half++: $\alpha$ and $\beta$ set to $0.7$ according to the Algorithm 1 for the generator and discriminator trained on CASIA-WebFace dataset. This is done for all $10,000$ random mixture of identities selected from $\mathbb{L}_s$

The results for this are shown in the Table 3, here as we observe on almost all of the benchmarks the $D^{aug}$ generated using $\alpha$ and $\beta$ values with higher $m^{total}$ are performing better.

Table 3: Effectiveness of our weighting procedure (W/ Half++) in comparison to (W/ Random) or when putting the conditions to 0.5 (W/ Half) and when setting the condition signal to 1 (W/ Full). Best in bold, second best, underlined.

| C Weight Method | $n^s$ | $n^r$ | B-1e-6 | B-1e-5 | C-1e-6 | C-1e-5 | TR1 | $m^{\text{total}}$ |
|---|---|---|---|---|---|---|---|---|
| W/ Half | ∼0.5M | 0 | 8.52±5.61 | 27.74±6.87 | 11.59±4.26 | 35.69±5.23 | 46.42±0.60 | 1.48 |
| W/ Full | ∼0.5M | 0 | 17.63±0.08 | 32.47±0.47 | 24.30±0.80 | 37.45±0.22 | 45.08±0.17 | 1.53 |
| W/ Random | ∼0.5M | 0 | 24.47±1.23 | 39.83±1.08 | 30.79±1.39 | 44.33±0.88 | **49.34±0.31** | N/A |
| W/ Half++ | ∼0.5M | 0 | **25.44±0.19** | **46.20±0.12** | **39.66±0.38** | **51.47±0.29** | 47.95±0.09 | **1.58** |

## 4.4 MIXING EFFECT

In Table 4, the effect of increasing the number of samples in our augmented dataset using $(\alpha, \beta) = (0.7, 0.7)$ weights is shown. On average, adding more classes (#Class) and samples per class (#Sample) improves the performance of the final discriminative model. The performance eventually decreases as more samples are added per class. We hypothesize that this is due to the similarity of images generated under the new conditions, $c$, when sampling $G(\mathbf{Z}, c)$ multiple times. This reduces the intra-class variability necessary for training an effective discriminator. We also observe that we should add an appropriate number of the augmentation dataset (*i.e.*, comparing 10k $\times$ 5 to without any augmentation) for the final performance to be better than the discriminator trained on the original dataset.

Table 4: Effect of mixing different numbers of classes (#Class) and samples per class (#Sample) with the original data, CASIA-WebFace. For TinyFace Rank-1 and Rank-5 accuracies are presented as TR1 and TR5 respectively.

| Syn #Class $\times$ #Sample | $n^r$ | B-1e-6 | B-1e-5 | C-1e-6 | C-1e-5 | TR1 | TR5 |
|---|---|---|---|---|---|---|---|
| 0 | 0.5M | 1.16±0.08 | 5.61±1.64 | 0.83±0.10 | 5.86±1.31 | 58.01±0.28 | 63.47±0.07 |
| Ours (5k $\times$ 5 ) | 0.5M | 0.85±0.06 | 5.60±0.84 | 0.65±0.08 | 6.70±0.97 | 58.19±0.20 | 63.48±0.01 |
| Ours (5k $\times$ 20) | 0.5M | 1.08±0.16 | 5.81±1.01 | 0.84±0.12 | 6.88±1.38 | 57.50±0.13 | 63.07±0.33 |
| Ours (5k $\times$ 50) | 0.5M | 0.63±0.23 | 4.56±0.41 | 0.46±0.10 | 6.55±0.35 | 57.39±0.20 | 62.55±0.11 |
| Ours (10K $\times$ 5) | 0.5M | 0.77±0.08 | 4.40±0.14 | 0.61±0.03 | 4.69±0.26 | 58.30±0.28 | 63.28±0.30 |
| Ours (10K $\times$ 20) | 0.5M | 1.29±0.01 | 8.21±1.38 | 1.43±0.22 | 9.67±1.01 | 58.01±0.50 | 63.00±0.71 |
| Ours (10K $\times$ 50) | 0.5M | 0.62±0.17 | 4.29±0.27 | 0.64±0.10 | 5.98±0.00 | 57.51±0.32 | 62.77±0.08 |

## 5 CONCLUSIONS

We have shown that by using a generator and discriminative model trained on a *single dataset*, we can generate an augmented dataset that will boost the performance of the discriminative model on several FR benchmarks, without relying on any auxiliary data or pre-trained model. We consistently outperformed the baseline discriminator model on various evaluation benchmarks, unlike other state-of-the-art models whose performance improvements were not consistent across evaluations.

**Future work.** Our method can be considered a general formulation of MixUp Zhang (2017) and CutMix Yun et al. (2019), but instead of cropping or blending images, we use a generator to create a new class. One interesting research direction would be to test if we can reformulate the margin losses used in FR to be compatible with the soft labels. Later by establishing a correlation between the target soft labels and the $c^*$, (*e.g.*, for $\alpha$ and $\beta$ set to 0.7 which increases the $m^{\text{total}}$ an obvious choice for soft target labels would be 0.5 and 0.5 for the corresponding source classes) one can study would it be beneficiary to treat the class as a soft-class or a new one.

**Reproducibility.** All code for the discriminative and generative models, along with the generated datasets and trained models, will be publicly available for reproducibility.

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

## A  FACE RECOGNITION BENCHMARKS

Here we present the same experiment setting as in Table 2 for IJB-B Whitelam et al. (2017) and IJB-C Maze et al. (2018) for more thresholds set by usual False Positive Rates (FPRs), respectively presented in Table 5 and Table 6. We can observe again that our generated images consistently improve the discriminator trained on the original dataset in both the benchmarks and all the FPR values.

Table 5: Comparison of the fully synthetic FR training, fully real FR training, and mixed FR training, when the models are evaluated against IJB-B with various FR thresholds. Here $n^s$ and $n^r$ depict the number of Synthetic and Real Images respectively and Aux depicts whether the method for generating the dataset using an auxiliary information network for generating their datasets

| Method/Data | Aux | $n^s$ | $n^r$ | B-1e-6 | B-1e-5 | B-1e-4 | B-1e-3 | B-0.01 | B-0.1 | Avg |
|---|---|---|---|---|---|---|---|---|---|---|
| DigiFace1M | N/A | 1.22M | 0 | 15.31±0.42 | 29.59±0.82 | 43.53±0.77 | 59.89±0.51 | 76.62±0.44 | 91.01±0.12 | 52.66±0.47 |
| RealDigiFace | Y | 1.20M | 0 | 21.37±0.59 | 39.14±0.40 | 52.61±0.70 | 67.68±0.73 | 81.30±0.56 | 93.15±0.17 | 59.21±0.52 |
| IDiff-face | Y | 1.2M | 0 | 26.84±2.03 | 50.08±0.48 | 64.58±0.32 | 77.19±0.41 | 88.27±0.15 | 95.94±0.05 | 67.15±0.50 |
| DCFace | Y | 0.5M | 0 | 29.74±2.25 | 57.55±0.76 | 73.00±0.39 | 83.87±0.28 | 92.29±0.17 | 97.34±0.06 | 72.30±0.65 |
| DCFace | Y | 1.2M | 0 | 22.48±4.35 | 47.84±6.10 | 73.20±2.53 | 86.11±0.59 | 93.55±0.16 | 97.56±0.06 | 70.12±2.28 |
| Auggen, D$^{aug}$ (Ours) | N | ∼0.6M | 0 | 32.67±1.17 | 51.52±0.69 | 67.77±0.83 | 80.24±0.50 | 90.30±0.03 | 96.74±0.03 | 69.87±0.52 |
| Auggen D$^{repro}$ (Ours) | N | ∼0.6M | 0 | 15.71±3.12 | 45.97±4.64 | 73.05±0.89 | 85.54±0.16 | 93.52±0.17 | 97.82±0.08 | 68.60±1.43 |
| CASIA-WebFace | N/A | 0 | ∼0.5M | 1.16±0.08 | 5.61±1.64 | 50.32±4.65 | 87.03±0.38 | 95.41±0.09 | 98.36±0.04 | 56.31±1.13 |
| IDiff-face | Y | 1.22M | ∼0.5M | 0.89±0.07 | 5.80±0.63 | 54.76±2.31 | 88.33±0.49 | 96.02±0.04 | 98.59±0.03 | 57.40±0.56 |
| DCFace | Y | 0.5M | ∼0.5M | 0.26±0.11 | 1.59±0.51 | 35.62±7.89 | 84.30±3.52 | 95.10±0.46 | 98.36±0.08 | 52.54±2.08 |
| Auggen, D$^{aug}$ | N | ∼0.2M | ∼0.5M | 1.29±0.01 | 8.21±1.38 | 57.12±4.32 | 87.98±0.50 | 95.31±0.25 | 98.45±0.02 | 58.06±1.07 |

Table 6: Comparison of the fully synthetic FR training, fully real FR training, and mixed FR training, when the models are evaluated against IJB-C with various FR thresholds. Here $n^s$ and $n^r$ depict the number of Synthetic and Real Images respectively and Aux depicts whether the method for generating the dataset using an auxiliary information network for generating their datasets

| Method/Data | Aux | $n^s$ | $n^r$ | C-1e-6 | C-1e-5 | C-1e-4 | C-1e-3 | C-0.01 | C-0.1 | Avg |
|---|---|---|---|---|---|---|---|---|---|---|
| DigiFace1M | N/A | 1.22M | 0 | 26.06±0.77 | 36.34±0.89 | 49.98±0.55 | 65.17±0.39 | 80.21±0.22 | 92.44±0.05 | 58.37±0.46 |
| RealDigiFace | Y | 1.20M | 0 | 36.18±0.19 | 45.55±0.55 | 58.63±0.59 | 72.06±0.90 | 84.77±0.59 | 94.57±0.19 | 65.29±0.50 |
| IDiff-face | Y | 1.2M | 0 | 41.75±1.04 | 51.93±0.89 | 65.01±0.63 | 78.25±0.39 | 89.41±0.19 | 96.55±0.05 | 70.48±0.47 |
| DCFace | Y | 0.5M | 0 | 51.64±1.55 | 64.58±1.01 | 76.98±0.74 | 86.90±0.38 | 93.90±0.07 | 97.82±0.01 | 78.64±0.63 |
| DCFace | Y | 1.2M | 0 | 35.27±10.78 | 58.22±7.50 | 77.51±2.89 | 88.86±0.69 | 94.81±0.09 | 98.06±0.06 | 75.46±3.65 |
| Auggen, D$^{aug}$ (Ours) | N | ∼0.6M | 0 | 47.74±0.47 | 58.07±0.48 | 71.61±0.50 | 82.87±0.32 | 92.03±0.04 | 97.37±0.04 | 74.95±0.31 |
| Auggen D$^{repro}$ (Ours) | N | ∼0.6M | 0 | 31.54±6.65 | 58.61±3.89 | 78.11±0.51 | 88.51±0.04 | 94.79±0.09 | 98.17±0.04 | 74.96±1.82 |
| CASIA-WebFace | N/A | 0 | ∼0.5M | 0.83±0.10 | 5.86±1.31 | 56.87±3.14 | 89.41±0.40 | 96.19±0.06 | 98.61±0.02 | 57.96±0.83 |
| IDiff-face | Y | 1.22M | ∼0.5M | 0.70±0.11 | 7.46±2.08 | 57.43±4.17 | 89.89±0.71 | 96.63±0.08 | 98.77±0.01 | 58.48±1.19 |
| DCFace | Y | 0.5M | ∼0.5M | 0.18±0.07 | 1.54±0.59 | 38.17±8.24 | 86.18±3.32 | 95.88±0.42 | 98.59±0.05 | 53.42±2.11 |
| Auggen, D$^{aug}$ | N | ∼0.2M | ∼0.5M | 1.43±0.22 | 9.67±1.01 | 61.75±3.48 | 90.00±0.44 | 96.17±0.19 | 98.64±0.01 | 59.61±0.81 |

## B  EXPERIMENT DETAILS

### B.1  DISCRIMINATOR TRAINING

In the Table 7 the most important parameters for training our discriminative models are presented.

### B.2  GENERATOR AND ITS TRAINING

We trained two sizes of generator namely small and medium as in Karras et al. (2024). The training of the small-sized generator took about 1 NVIDIA H100 GPU day for the generator to see 805M images with a batch size of 2048. For reaching the same number of training images for the medium-sized generator, took about 2 days with a batch size of 1024. We used an Exponential Moving Average (EMA) length of 10%. As observed in literature Nichol & Dhariwal (2021), the EMA of model weights plays a crucial role in the output quality of the Image Generators.

For sampling our models we did **not** employ any Classifier Free Guidance (CFG) Ho & Salimans (2021).

### B.3 Table Details

For the Table 3 we conditioned a medium-sized generator which trained till it saw 805M images. The conditions were set according to the four sets of values of the $\alpha$ and $\beta$. This is done for a fixed identity combination from the $\mathbb{L}_s$ for all of them. Later for each of these new conditions $\boldsymbol{c}^*$ we generated 50 images. All other tables were reported from a medium-sized generator when they saw 335M training samples.

Table 7: Details of the Discriminator and its Training

| Parameter Name | Discriminator Type 1 | Discriminator Type 2 |
|---|---|---|
| Network type | ResNet 50 | ResNet 50 |
| Marin Loss | AdaFace | AdaFace |
| Batch Size | 192 | 512 |
| GPU Number | 4 | 1 |
| Gradient Acc Step | 1 (For every training step ) | N/A |
| GPU Type | Nvidia RTX 3090 Ti | Nvidia H100 |
| Precision of Floating Point Operations | High | High |
| Matrix Multiplication Precision | High | High |
| Optimizer Type | SGD | SGD |
| Momentum | 0.9 | 0.9 |
| Weight Decay | 0.0005 | 0.0005 |
| Learning Rate | 0.1 | 0.1 |
| WarmUp Epoch | 1 | 1 |
| Number of Epochs | 26 | 26 |
| LR Scheduler | Step | Step |
| LR Milestones | $[12, 24, 26]$ | $[12, 24, 26]$ |
| LR Lambda | 0.1 | 0.1 |
| Input Dimension | $112 \times 112$ | $112 \times 112$ |
| Input Type | RGB images | RGB Images |
| Output Dimension | 512 | 512 |
| Seed | 41,2048,10 (In some models) | 41,2048 |

## C Downstream Performance vs Metrics in Generative Models

In this section, we examine whether there is a correlation between common metrics for evaluating generative models and the discriminator's performance when trained on our augmented dataset. We studied the FD Heusel et al. (2017) Precision/Recall Sajjadi et al. (2018); Kynkäänniemi et al. (2019) and Coverage Naeem et al. (2020) which is usually used to quantify the performance of the Generative Models. Calculation of these metrics requires the projection of the images into meaningful feature spaces. For feature extraction, we consider two backbones, Inception-V3 Szegedy et al. (2016) and DINOv2 Oquab et al. (2023) which are shown effective for evaluating diffusion models Stein et al. (2023). Both these models were trained using the ImageNet Russakovsky et al. (2015) in a supervised and semi-supervised manner respectively. Experiments were performed by randomly selecting $100,000$ images of both CASIA-WebFace (as the source distribution) and our generated images by value of $\alpha$ and $\beta$ using Algorithm 1 (*i.e.*, the same settings as presented in the section 4). We are reporting four versions of our generated augmentation using a medium-sized generator when it sees 184M, 335M, 603M, and 805M training samples (M for Million). For each of the classes generated from these models we selected 20 samples, based on the observation in Table 4. Later by mixing the selected images with the original CASIA-WebFace we train FR for each of them and reporting the average accuracies for different thresholds in the IJB-C (*i.e.*, similar to last column in the Table 6). Figure 5 and Figure 6 are showing mentioned metrics for Inception-V3 and DINOv2 feature extractor respectively. We observe no clear correlation between the metrics used to evaluate generative models and the performance of a downstream task. This holds when we are

augmenting the dataset for training the generator and discriminator with the original dataset. This highlights the need to develop new evaluation metrics.

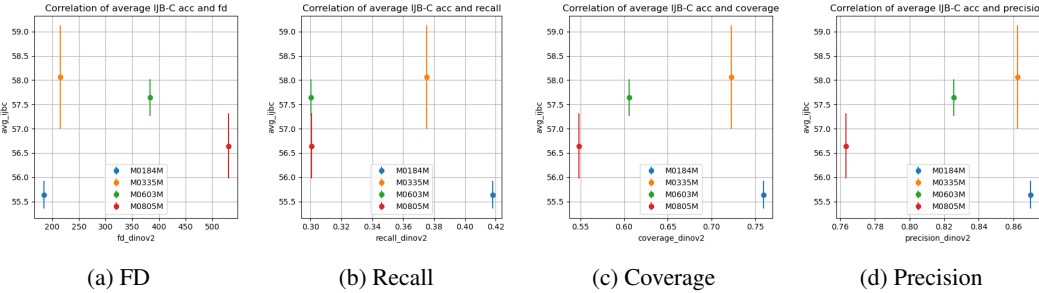

(a) FD  (b) Recall  (c) Coverage  (d) Precision

Figure 5: Correlation between the FD, Recall, Coverage, and Precision for the generated dataset by comparing it with the features of CASIA-WebFace using DINOv2 extractor.

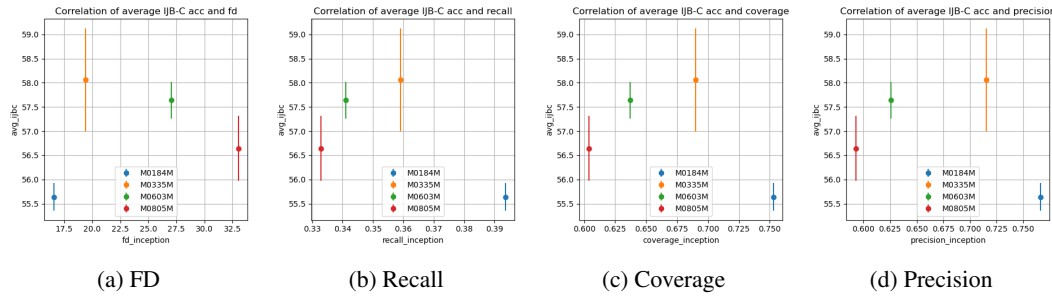

(a) FD  (b) Recall  (c) Coverage  (d) Precision

Figure 6: Correlation between the FD, Recall, Coverage, and Precision for the generated dataset by comparing it with the features of CASIA-WebFace using Inception-v3 extractor.

## D    GENERATED IMAGES

In the following figures, you can find more examples of generated images for Small and Medium-sized generators and also trained for more steps. By comparing Figure 7 (generated result from a small-sized generator trained when it sees 335M images, **S335M**), Figure 8 (**M335M**) and Figure 9 (**M805M**) we generally observe that larger generators are producing better images, but training for more steps does not necessarily translate to better image quality.

918
919
920
921
922
923
924
925
926
927
928
929
930
931
932
933
934
935
936
937
938
939
940
941
942
943
944
945
946
947
948
949
950
951
952
953
954
955
956
957
958
959
960
961
962
963
964
965
966
967
968
969
970
971

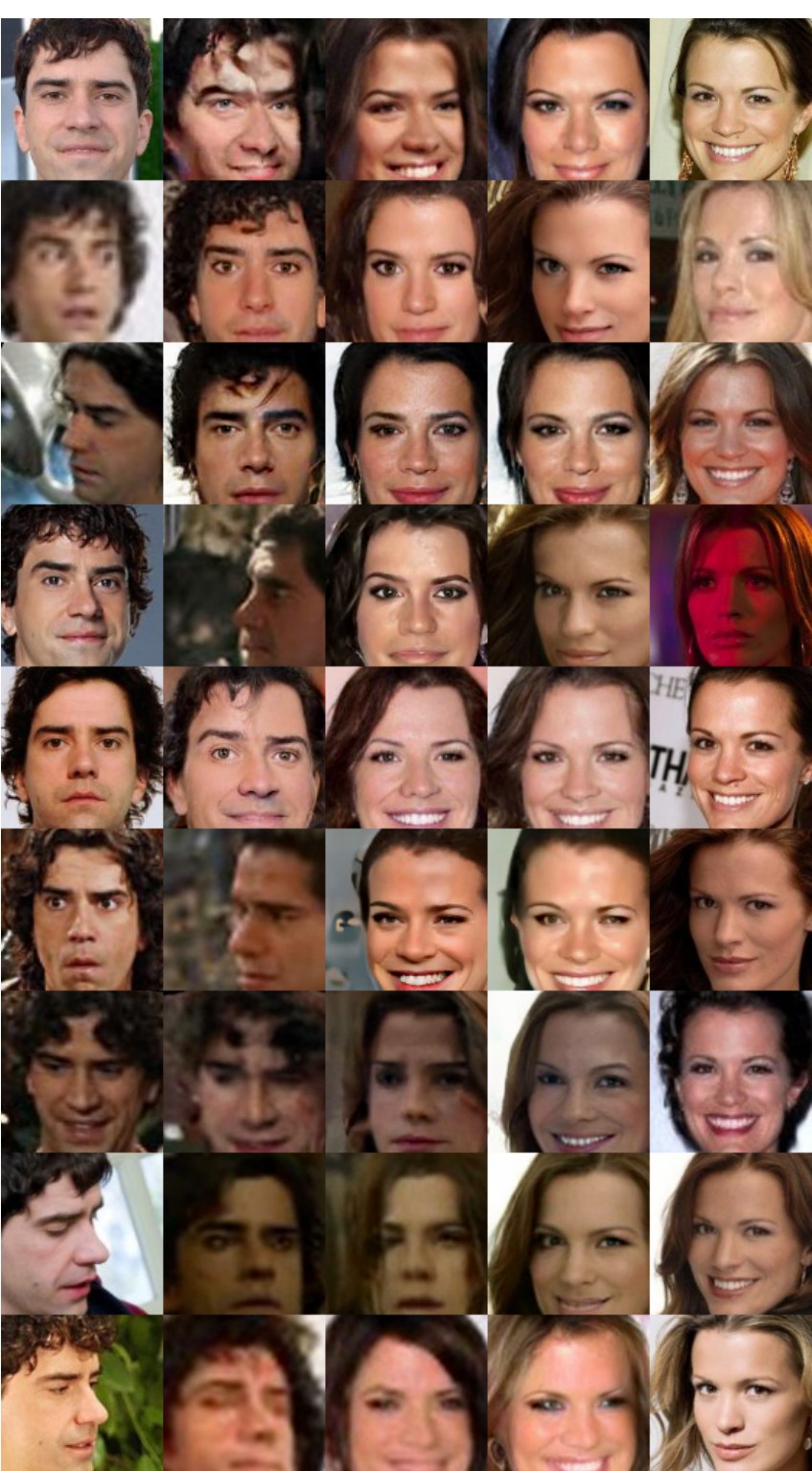

Figure 7: Samll-sized generator trained till it sees 335M images. From left to right, the first column is variations of a random ID, 1, in the, $D^{orig}$, the second column is the recreation of the same ID in the first column using the generator when we put the conditions to 1, The last two columns are the same but for different IDs and the middle column representing the interpolated new identity.

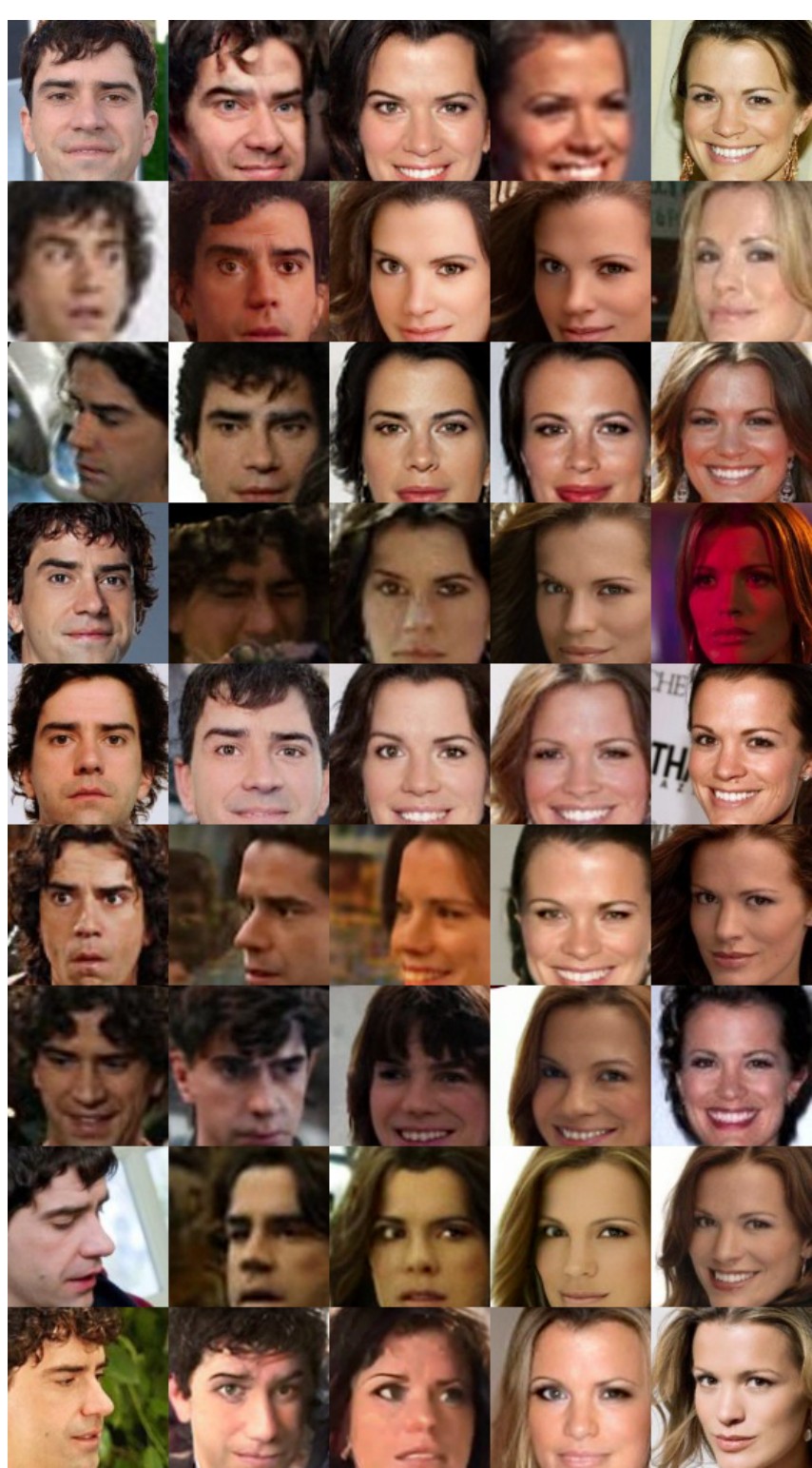

Figure 8: Medium-sized generator trained till it sees 335M images. From left to right, the first column is variations of a random ID, 1, in the, D$^{orig}$, the second column is the recreation of the same ID in the first column using the generator when we put the conditions to 1, The last two columns are the same but for different IDs and the middle column representing the interpolated new identity.

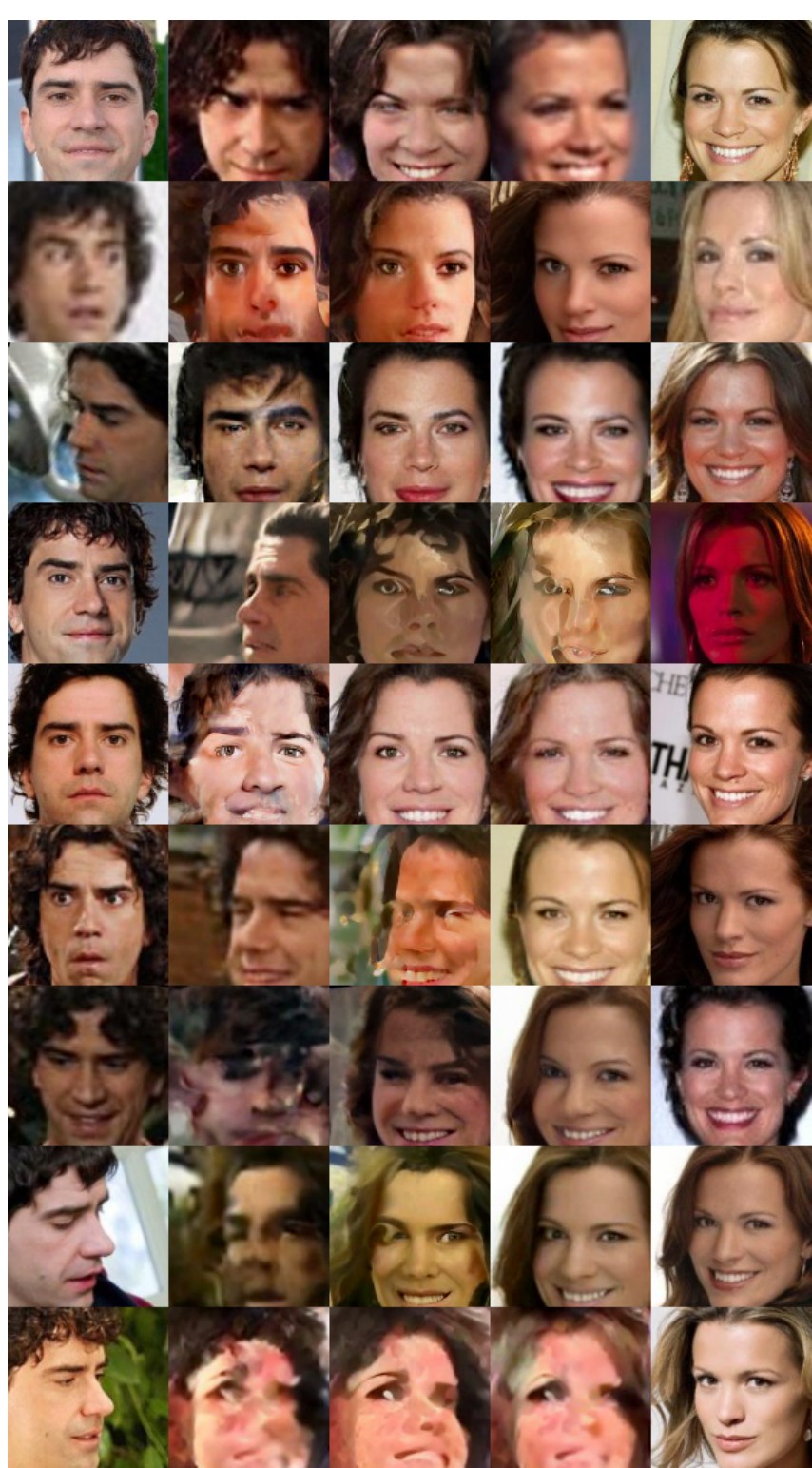

Figure 9: Medium-sized generator trained till it sees 805M images. From left to right, the first column is variations of a random ID, 1, in the, $D^{orig}$, the second column is the recreation of the same ID in the first column using the generator when we put the conditions to 1, The last two columns are the same but for different IDs and the middle column representing the interpolated new identity.

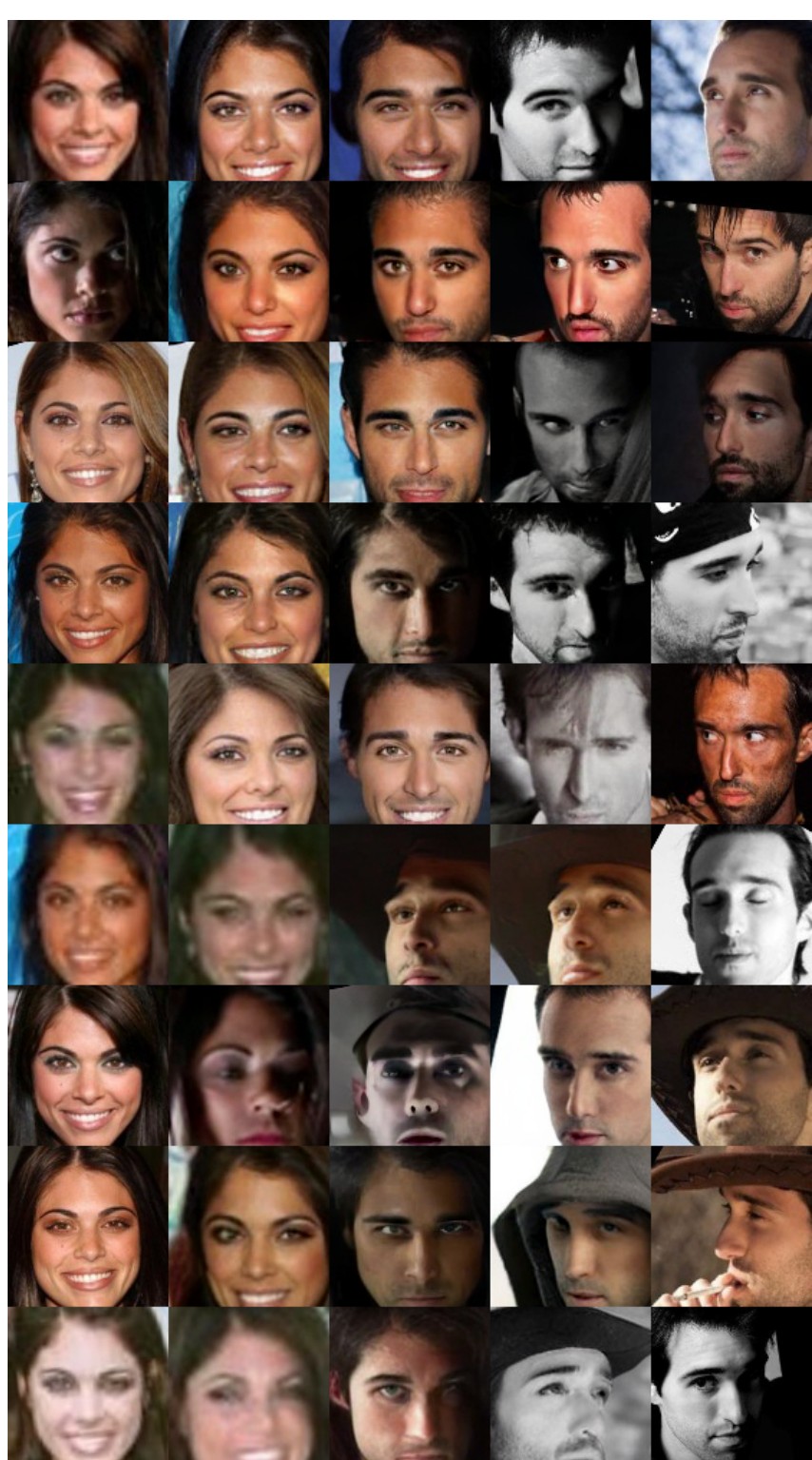

Figure 10: Medium-sized generator trained for till it sees 335M images for different IDs. From left to right, the first column is variations of a random ID, 1, in the, $D^{orig}$, the second column is the recreation of the same ID in the first column using the generator when we put the conditions to 1, The last two columns are the same but for different IDs and the middle column representing the interpolated new identity.

