# OpenReview forum: "AugGen: Generative Synthetic Augmentation Can Boost Face Recognition"
_ICLR.cc/2025/Conference — Submitted to ICLR 2025_

### Official Review · Reviewer_ujs9 · 2024-11-03

**Soundness:** 3
**Presentation:** 3
**Contribution:** 2
**Rating:** 5
**Confidence:** 4

**Summary:**

This paper addresses the challenge of training face recognition models with synthetic data to mitigate privacy and ethical concerns while enhancing model performance. The proposed approach introduces a new sampling method for learning a face generator. During training, both real and synthetic face data are used to train the face recognition model. Experiments are conducted on two small datasets to demonstrate the effectiveness of the proposed method.

**Strengths:**

- This paper addresses an important challenge in face recognition by exploring the use of synthetic data for model training.

- Experimental results highlight the effectiveness of integrating real and synthetic data for training face recognition models.

**Weaknesses:**

- The proposed method continues to depend on large-scale real-world face images for training, which diminishes its impact on addressing privacy and ethical concerns.

- The employed frameworks are similar to several existing methods, but references are missing. For example, [1] Synface: face recognition with synthetic data, 2021; [2] SFace: Privacy-friendly and accurate face recognition using synthetic data, 2022

- Figures 1 and 2 contain excessive overlapping content that could be reduced or eliminated for clarity.

**Questions:**

See weakness.

---

> ### Author Response · Authors · 2024-11-22
>
> We thank the reviewer for the time and thoughtful feedback to improve our work.
>
> * As suggested, we **will merge** Figures 1 and 3 in the final version for clarity and as we need to save space to include further experiments.
>
> * Please note we are trying to **alleviate** the need for large-scale datasets, **unlike** current approaches that we compare our method does not require any pre-trained FR which on its own requires large-scale datasets (**we are the first** to do so).
>
> * Due to space constraints, we focused on state-of-the-art methods but will include SFace and SynFace in the background section (2). Notably, DCFace and IDiffFace, which we primarily compare with, outperform SynFace and SFace.
>
>     Additionally, please note that:
>
>     * **SynFace**: Built on DiscoFaceGAN with auxiliary networks and 3DMM. In contrast, our method mixes in the generator’s condition space with uniquely set parameters (not interpolations, based on Algorithm.1 ). SynFace interpolates in **input** and **label** space like the original Mixup paper (Eq.3, Sec. 3.4).
>
>      * **SFace**: Uses class conditional StyleGAN-ADA but does **not mix in condition space** as found by our Algorithm. 1, instead of generating images with the same conditions (Sec. II-A) and employing a combined output loss.
>
> We appreciate the reviewer’s insights and will reflect these points in the manuscript. Please note that to **the best of our knowledge**, we are the **first** to do conditioning mixing as described in Alg.1.
>
> Finally, we kindly ask the reviewer to check the **additional experiments** on separate datasets (WebFace10K)  included during the rebuttal, further validating our findings. Also, we added an explanation on why we think our method works in the **general response**.

---

> ### Author Response · Authors · 2024-11-26
>
> Dear Reviewer ujs9
>
> In response to your comments, we have addressed the following:
>
> * Outlined **what distinguishes** our method from the references you mentioned.
> * Demonstrated the utility of our approach (**beyond architecture level improvements**), even when relying on the FR dataset, while showcasing its clear performance boost.
>
> Please let us know if further clarifications are needed.

---

### Official Review · Reviewer_1S4a · 2024-11-03

**Soundness:** 2
**Presentation:** 2
**Contribution:** 2
**Rating:** 5
**Confidence:** 4

**Summary:**

The paper aims to address privacy and ethical concerns associated with large-scale real datasets while improving the performance of face recognition models using the synthetic data. It employs one single dataset to train both generator and discriminator to eliminate the need for auxiliary data or pre-trained models. Finally, the authors claim that training on a mix of real and synthetic data improves performance on discriminative tasks compared to training on real data alone.

**Strengths:**

1. The use of synthetic data alleviates privacy and ethical concerns associated with large-scale real datasets.
2. The method does not require additional data or pre-trained models for the generator.

**Weaknesses:**

1. Using synthetic data to avoid privacy concerns or to enhance the performance of face recognition systems has been explored in several prior methods. This paper claims a distinct advantage in that it does not require additional data or pre-trained models, as it relies on a single dataset for training both the generator and the discriminator.
    - Training a new generator using a dataset of 805M in size incurs substantial costs,
    - The above data appears to differ from that used to train face recognition models, such as CASIA-WebFace, which contains only 0.5M samples. This may impliy the use of additional data, so it would be more illustrative if the authors can explain these details.
    - From the perspective of using a single training dataset, the generator aims to learn the distribution of the input dataset. When sampling, it produces an image that aligns with this distribution, which can be regarded as interpolating the input face images at the identity level. This process is expected to increase the diversity of the original data, thereby contributing to performance improvements. Utilizing other pre-trained models introduces additional dataset diversity to the original data, potentially resulting in even greater gains.

2. The paper lacks comparisons with related methods such as Synface and SFace. Synface, for instance, investigates the integration of synthetic and real data during training, and its identity mixup technique involves combining two distinct identities using varying coefficients, a process similar to Equation 4 in this paper.

3. There are areas where the writing could be refined. For example:
    - On line 65, the text mentions "M_mix," whereas Figure 1 labels it as "M_new"; Figure 2, however, uses "M_mix."
    - On line 164, it states "k pairs of image and label," yet the listed pairs of x and y range from 0 to k, indicating there are actually k+1 pairs.

**Questions:**

Please refer to the **Weaknesses**.

---

> ### Author Response · Authors · 2024-11-22
>
> We sincerely thank the reviewer for their time and valuable feedback.
>
> * Both the generator and discriminator were trained solely on CASIA-WebFace. The numbers (e.g., 335M, 805M) indicate how many times the generator processed the 0.5M CASIA images with different noise scales, **with 335M corresponding to ~700 epochs**. Diffusion model training involves creating and denoising noisy image versions, and this reporting aligns with standard practices (e.g., EDM, EDM2, RIN). We will **clarify** this in the manuscript.
>
> * Please refer to the general comment on why we think our **method is boosting the performance of the discriminator.**
>
> * Regarding comparisons, we focused on state-of-the-art methods due to space constraints but will include SFace and SynFace in the related work section:
>
>     * **SynFace**: Built on DiscoFaceGAN with auxiliary networks and 3DMM. In contrast, our method mixes in the generator’s condition space with uniquely set parameters (not interpolations, based on Algorithm.1 ). SynFace interpolates in **input** and **label space** like the original Mixup paper (Eq.3, Sec. 3.4).
>
>     * **SFace**: Uses class conditional StyleGAN-ADA but does not mix in condition space as found by our Algorithm. 1, instead of generating images with the same conditions (Sec. II-A) and employing a combined output loss.
> * We thank the reviewer for spotting these typos and will try to clarify this in the manuscript.
>
>
> Finally, we kindly ask the reviewer to check the **additional experiments** on separate datasets (WebFace10K) **in the general response** included during the rebuttal, further validating our findings.

---

> ### Author Response · Authors · 2024-11-26
>
> Dear Reviewer 1S4a,
>
> We sincerely appreciate your time and effort in reviewing our paper.
>
> In response to your feedback, we have addressed the following:
>
> * Clarified that the same dataset was used and demonstrated consistent improvements by redoing experiments on a **new dataset**.
> * Provided an **explanation for why our method works**, supported by experiments on **Separation of Classes** and **Compactness** (Please refer to general response).
> * Highlighted **what distinguishes** our simple and effective method from the references you suggested.
>
> Please let us know if you have any further questions or need additional clarifications. We would be happy to assist.

---

### Official Review · Reviewer_1Ykd · 2024-11-03

**Soundness:** 2
**Presentation:** 2
**Contribution:** 2
**Rating:** 3
**Confidence:** 4

**Summary:**

This paper addresses the problem of the need for large-scale datasets to train deep networks by augmenting the original dataset with synthetic data. To this end, the paper trains a conditional generative model to synthesize new examples. The paper argues that, beyond generating new examples from existing classes, it is also beneficial to generate examples of new classes by interpolating the conditional vectors between two different classes.

Both the generative and discriminative models are trained on CASIA-Web datasets, and their performances are compared with models trained on DigiFace 1M, Real DigiFace, DCFace, etc.

**Strengths:**

The paper addresses the important issue of the need for a large-scale training set.

The paper is generally easy to follow.

**Weaknesses:**

A concern is the novelty of the method. The idea of interpolating conditional vectors to train a generative model is somewhat incremental. There are multiple works on GANs for face attribute manipulation using continuous features [ref1] or differences in one-hot vectors [ref2].


Additionally, the paper mostly relies on empirical values to justify the idea, with no detailed explanation of why the proposed method of interpolating conditional vectors to synthesize new identities works. Conditioning on a large one-hot vector (e.g., 10,000 for a dataset with 10K identities in the training set) can also be challenging to train, but this issue is not discussed.

The method is evaluated only on CASIA-Web. It would be beneficial to evaluate on at least one more dataset if relying on empirical values to validate the idea.

The paper could have dedicated more space to providing detailed insights into the method rather than spending too much on the basics of generative and discriminative models.


[ref1] Bhattarai, B., & Kim, T. K. (2020). Inducing optimal attribute representations for conditional GANs. In Computer Vision–ECCV 2020: 16th European Conference, Glasgow, UK, August 23–28, 2020, Proceedings, Part VII 16 (pp. 69-85). Springer International Publishing.

[ref2] Liu, M., Ding, Y., Xia, M., Liu, X., Ding, E., Zuo, W., & Wen, S. (2019). Stgan: A unified selective transfer network for arbitrary image attribute editing. In Proceedings of the IEEE/CVF conference on computer vision and pattern recognition (pp. 3673-3682).

**Questions:**

A clear articulation of the novelty of methodology, a detailed explanation of why the method works, and experiments on yet another benchmark can change my opinion about the paper.

---

> ### Author Response · Authors · 2024-11-22
>
> We appreciate the time and effort the reviewer put into reviewing our manuscript:
>
> About the references:
>
> - **[ref1] Inducing Optimal Attribute Representations for Conditional GANs:** This method focuses on attribute editing rather than dataset augmentation for boosting FR performance. It uses networks such as Graph Convolutional Networks and co-occurrence matrices, whereas our approach employs a simple class-conditional generator for augmentation without additional complexities.
>
> - **[ref2] STGAN:** This method introduces attribute editing (e.g., adding glasses) by using attribute difference codes in its loss function. Unlike our approach, it does not generate augmentation datasets by mixing identities in the generator’s condition space. Our method relies on a pre-trained class-conditional generator without requiring optimization of the generator itself.
>
> Neither of these methods **incorporates offline condition-space mixing as we do**. Using Algorithm 1, our mixing parameters are calculated in a few forward passes without complex optimization, making it practical even for state-of-the-art diffusion models.
>
> ## Additional Clarifications:
>
> * We tested both torch’s nn.Embedding layer and one-hot condition vectors. The one-hot vectors are projected into a 768-dimensional condition space by a Magnitude Preserving Linear Layer following EDM2 observations. We will release the dataset and code to facilitate further adoption and we propose to add more details on the conditioning mechanism in the appendix.
> * As requested, **we conducted experiments with another dataset**, a subset of WebFace4M (WebFace10K). Please check the **general response**, briefly, we trained a generator from scratch, sampled 10K identity mixes, and augmented this with the original dataset, WebFace10K in this case. This experiment confirmed similar performance improvements, as described in the opening section.
> * Lastly,  as requested, **we added an explanation of why we think our methodology works** and we will incorporate this into the Methodology section of the paper.

---

> ### Author Response · Authors · 2024-11-26
>
> Dear Reviewer 1Ykd,
>
> Thank you for your valuable feedback and for taking the time to review our paper.
>
> As per your suggestions, we have addressed the following :
>
> * Clearly articulated what **sets** our simple yet effective method **apart**.
> * Provided an explanation for **why our method works**, **supported by experiments** (Separation of Classes and Compactness).
> * Applied our method to an **additional dataset**, demonstrating consistent effectiveness.
>
> Please let us know if you have any further questions or require additional clarifications. We are happy to provide further insights as needed.

---

### Official Review · Reviewer_y7jg · 2024-11-04

**Soundness:** 2
**Presentation:** 2
**Contribution:** 1
**Rating:** 5
**Confidence:** 3

**Summary:**

The paper proposes training face recognition models on a mix of real data and synthetic data generated with a diffusion model. Given a dataset of faces and identites, the authors learn a class-conditioned generative model, sample from it by interpolating between one-hot class encodings, and mix these samples with real data when training a discriminative model. They demonstrate this procedure leads to improvement with respect to a model trained solely on real data.

**Strengths:**

The experimental evaluation is the main strength of the paper. Authors test their method on a few benchmarks over multiple seeds and compare to appropriate baselines.

**Weaknesses:**

The main weakness of the paper is that there is little theoretical justification for what authors are trying to achieve. For synthetic data to significantly improve the performance of a machine learning model, it has to introduce new information. This is the case when the data comes from a rendering pipeline, but expecting significant gains by training a discriminative model on the outputs of a generative model of the same distribution goes against the data processing inequality. It is possible that including generated samples helps in extremely data-impoverished or class-imbalanced scenarios, or improves the training dynamics of the discriminative model, but it cannot lead to major improvements. Indeed, the results in Table 1 show that the observed differences between a model trained on the real data and one trained on the outputs of a generative model are not statistically significant.

The proposed method has been explored before, e.g. by Hong et al. 2023 (Enhancing Classification Accuracy on Limited Data via Unconditional GAN), Besnier et al. 2019 (This Dataset Does Not Exist: Training Models from Generated Images), ane Lomurno et al. 2024 (Stable Diffusion Dataset Generation for Downstream Classification Tasks).

Presentation is a significant weakness of the submission. The descriptions are hard to follow because of convoluted wording, stylistic mistakes, and incorrect punctuation. The manuscript would benefit significantly from a thorough proofread. Frequent explanations of basic concepts (like a random seed) distract from the main ideas of the paper.

**Questions:**

Have you tried measuring the relationship between the size of the synthetic dataset and downstream metrics?

**Details Of Ethics Concerns:**

Direct applications in face recognition without thorough discussions of ethical implications.

---

> ### Author Response · Authors · 2024-11-22
>
> We appreciate the reviewer’s time and effort in reviewing our work.
>
>
> To address the reviewers' concerns, we have added an explanation and illustration to justify our method, which will be included in the final manuscript. We would like to clarify that our method **does not generate new information but produces images that enhance the discriminator's capabilities**, aligned with margin losses commonly employed in FR tasks. As requested, we performed an additional experiment on a **less class-imbalanced dataset** and observed similar improvements. We kindly ask the reviewer to check the **general response**.
> We felt it necessary to describe our method from the fundamentals, as practices such as seed changes, while impactful (please look at the variances ~0.1–1%), are not commonly emphasized in the FR community. We hope this explanation will encourage their broader adoption. Additionally, our results **demonstrate statistically significant** improvements, with up to **a 5% performance boost** on challenging IJB-B and IJB-C benchmarks.
> Regarding the references mentioned by the reviewer:
> - Hong et al., 2023: This work uses an unconditional GAN and cycle consistency approach, differing fundamentally from our simpler class-conditional generator. Their results on CIFAR (a smaller dataset) contrast with our improvements demonstrated across eight benchmarks and 10,000 classes.
> - This Dataset Does Not Exist, 2019: This method optimizes in the z-space with hard sampling, whereas our approach **mixes in the condition space** (Algorithm 1). This makes our method more practical for diffusion models which optimizing with respect to input noise is challenging.
> - Stable Diffusion Dataset Generation, 2024: This approach requires fine-tuning Stable Diffusion 2.0 for datasets like CIFAR and MNIST. Our study **avoids relying on pre-trained networks**, emphasizing fundamental exploration instead.
> We sincerely thank the reviewers for their thoughtful feedback and will include these references in the related work section, highlighting their distinctions from our approach.
>
>
>
> ## Answer to Questions:
>
> Indeed we explored various mixtures of synthetic and real images, including different numbers of generated classes and samples per class, as detailed in Table 5 of the paper.
>
>
> ## About Ethics:
>
> To train an FR system, datasets with face images across various identities are necessary. We primarily use CASIA-WebFace, a widely adopted dataset for synthetic FR generation in recent works such as DCFace (CVPR 2023) and ID³ (NeurIPS 2024).

---

> ### Comment · Reviewer_y7jg · 2024-11-22
> **Response to Authors**
>
> Thank you for the thorough response, additional experiments, and a more detailed explanation of the method.
>
> Regarding the illustration of your method, I am not convinced that introducing the additional class M-N necessarily increases the cosine distance between M and N. By this logic, introducing more classes to a margin-loss discriminator would always lead to better separation between existing classes. Is this something you verify in your experiments by comparing e.g. distances between class centroids in the embedding space?
>
> Regarding the results on LFW, CFP-FP, CPLFW, AgeDB, CALFW, and IJB-C, could you help me understand the difference between AugGen, $D^{aug}$ and AugGen, $D^{aug}$ (Ours)? Is the former trained on CASIA, and the latter on WebFace? Why are the gains not materializing for AugGen (Ours) CASIA in the first table?
>
> Overall, I have to concede that the method does lead to improvements in some cases. However, I think the lack of theoretical grounding and a convincing explanation of why (and when) that is the case makes it hard to recommend acceptance. I raised my score to reflect the new experimental insights.

---

> ### Author Response · Authors · 2024-11-23
>
> We sincerely thank the reviewer for their follow-up and interest in our work.
>
> * Experimentally, we observe that models performing well with **AugGen better reproduce** $D^{\mathrm{orig}}$ (i.e., $D^{\mathrm{repro}}$), evidenced by lower $FD_{\mathrm{dinov2}}$ and $FD_{\mathrm{discriminator}}$. Here, FD (Frechet Distance) how close the distribution of the reproduced set is to the original data, with subscripts indicating the extractor used. We propose including these observations in the final manuscript, alongside our experiments linking $D^{\mathrm{auggen}}$ to final performance (Please refer to appendix for the current version of this study). Given the complexity of generative models, establishing a theoretical link to the discriminator's capabilities remains challenging and is left for future work. To our knowledge, we are the first to address this problem in this manner. Notably, even MixUp (introduced in 2017) lacks a clear theoretical explanation for its effectiveness to date. For reference, see:
>     - https://arxiv.org/pdf/2010.04819
>     - https://linkinghub.elsevier.com/retrieve/pii/S0952197623019759
>
>
> * We apologize for the typo, corrected in the rebuttal table and to be clarified in the final manuscript. **AugGen from Dataset** refers to augmentations generated solely using WebFace10K or CASIA-WebFace. While IR-101 outperforms IR-50 on CASIA-WebFace on LFW, CFP-FP, ... (first table), our method surpasses CASIA-WebFace IR-50 using the same IR-50 backbone for AugGen training. We argue that **enhancing discriminator capabilities via generative models can rival architectural improvements**. Model trained on AugGen using IR-50 outperforms even the model trained on the solely original dataset using IR-101 by up to 4% on IJB-B and IJB-C and **consistently** surpasses IR-50 models trained only on the original dataset across **all** studied benchmarks like LFW, CFP-FP, and CPLFW. This finding is a valuable contribution to the community.
>
>
> ## Verification of Hypothesis:
>
> Here, we validate the hypothesis in two ways:
> - Calculating the average inter-class Similarity between mixed classes before and after applying AugGen samples, **separation of classes**).
> -  Average intra-class similarity and standard deviation, **Compactness of samples within each class**.
>
> ### Separation of Classes
> We verified this hypothesis experimentally. We compared two models: $F_{\mathrm{AugGen}}$, trained on the Original Dataset + AugGen, and $F_{\mathrm{Before}}$ (baseline), trained only on the Original Dataset. For each AugGen mix (e.g., M-N mix), we calculated the similarity of the mixed classes (M and N) using both models across 10,000 classes, then averaged the absolute similarities. Our hypothesis is confirmed if $F_{\mathrm{AugGen}}$ produces lower average similarity (corresponding to higher $\theta_{\mathrm{ours}}$, as shown in the figure). The results support this:
>
> | AugGen (Ours) | Original (baseline) |
> | -----  | -------- |
> | 0.06645628 | 0.067214645 |
>
> [Updated, finished averaging over all the dataset]
>
> ### Compactness
> Like previous section, we used the $F_{\mathrm{AugGen}}$ and $F_{\mathrm{Before}}$ to extract embeddings for Mix classes (like M and N) for all mixed classes ($\sim$ 10K). Here we are reporting the per class average similarity within each class (every two different samples of the same class) and averaged it over all the mix classes. We do the same and reporting `std` for each class and averaged it over all mix classes:
>
> | Metric | AugGen (Ours) | Original (baseline) |
> | ------------ | ------------ | -----------|
> | Avg `Intra Sim` |  0.54911   | 0.49065  |
> | Avg `std`       | 0.12807  | 0.13499 |
>
> Here we are demonstrating that the compactness (please refer to how the ovals are more compact in the illustration ) of the embeddings are increased after applying our AugGen samples, further boosting the effectiveness of the margin losses.
>
> This finding validates the **core idea** driving our work.
>
> We also added this section in the general response as we find your comment insightful for the `verification of our hypothesis` for other reviewers.

---

> > ### Author Response · Authors · 2024-11-28
> >
> > Dear Reviewer y7jg,
> >
> > Thank you for your time and valuable feedback! Based on your suggestions (and those of other reviewers), we have:
> >
> > - Emphasized what **sets** our simple yet effective method **apart**.
> > - Explained **why our method works**, supported by experiments on **Separation of Classes** and **Compactness**.
> > - Tested our method on an **additional dataset**, demonstrating consistent effectiveness.
> >
> > We’re happy to provide further clarification if needed.

---

> ### Author Response · Authors · 2024-12-02
>
> Dear Reviewer y7jg,
>
> We have further verified the question you raised about our explanation:
> "Introducing more classes to a margin-loss discriminator improves separation between existing classes."
>
> For details, please see the new comment in the general response: Adding More AugGen augmentation makes the model better.
>
> Thank you for your feedback.

---

### Author Response · Authors · 2024-11-22
**Experiment on Another Dataset**

# Experiment on Another Dataset:
In our paper, we used the CASIA-WebFace dataset as it is widely used by recent work e.g. DCFace (CVPR 2023) and ID³ (NeurIPS 2024). Nevertheless, as suggested we evaluated our method on another dataset: WebFace4M. A further illustration of results is available at the following **anonymous** link, which yields observations similar to the original evaluations.
https://docs.google.com/document/d/e/2PACX-1vQcpzqHf-YThsUTENRD3ZCzLWGQpePODdLLaCUbdIPX0ayhJO_rbPVhXsuIX7rcVDhFxLxpuGypjr8x/pub
–
# Experimental details:
We use a subset of WebFace4M, selecting 10K classes with 10–20 samples per class (~160K images), which we refer to as WebFace10K (wf10k). We trained an IR50 discriminator and an EDM2-small generator, applying Algorithm 1. The number of images per class is K=5 and the values for $m^{\mathrm{total}}$ (similarly to Table 3 in the paper) are:

* 0.50-0.50: 0.6068
* 0.70-0.70: 0.7256
* **0.80-0.80: 0.7390**
* 1.00-1.00: 0.7230

Using these results, we randomly generated an augmented dataset with class mixes based on α=0.8 and β=0.8,  [367-373], confirming our previous findings on the effectiveness of our method.

Here are the results on the benchmarks reported in the paper, we also added the results on the original datasets (CASIA and WebFace10K) by a discriminator trained using the **IR101** network (parameterized more) denoted by $\textcolor{blue}{\mathbf{\dagger}}$. This ensures that the **observed improvement is as significant as the network architecture itself** since the methods using a **mix of real and AugGen data are trained exclusively on the less parameterized IR50 backbone**.


## Results on LFW, CFP-FP, CPLFW, AgeDB, CALFW
| Method | Aux | $n^{s}$ | $n^{r}$ | LFW | CFP-FP | CPLFW | AgeDB | CALFW | Avg |
| ---- |  ---- |  ---- |  ---- |  ---- |  ---- |  ---- |  ---- |  ---- |  ---- |
| CASIA-WebFace IR50 | N/A | 0   | $\sim$0.5M | 99.21±0.18 | 87.85±1.72 | 95.69±1.16 | 92.78±0.47 | 92.71±0.96 | 93.65±0.89  |
|CAISA-WebFace IR101 $^\dagger$ | N/A | 0 | $\sim$0.5M | 99.45±0.05 | 89.92±0.12 | 97.06±0.06 | 93.54±0.02 | 94.33±0.13 | 94.86±0.07  |
| AugGen from CASIA, IR50|  $\textcolor{teal}{N}$ | 0.2M | $\sim$0.16M | 99.41±0.08 | 89.32±0.02 | 96.41±0.09 | 93.13±0.03 | 93.63±0.15 | 94.38±0.00  |
| ---- |  ---- |  ---- |  ---- |  ---- |  ---- |  ---- |  ---- |  ---- |  ---- |
|WebFace10K IR50 | N/A | 0 |  $\sim$0.16M  | 99.08±0.13 | 87.99±0.45 | 93.95±0.59 | 92.75±0.20 | 90.78±0.79 | 92.91±0.42  |
|WebFace10K IR101 $^\dagger$  | N/A | 0 | $\sim$0.16M  | 98.97±0.11 | 87.54±0.06 | 93.40±0.01 | 92.55±0.02 | 90.01±0.04 | 92.50±0.02 |
| AugGen from WegFace10K IR50 | $\textcolor{teal}{N}$ | 0.2M | $\sim$0.16M | 99.33±0.06 | 88.73±0.32 | 95.20±0.14 | 93.42±0.11 | 91.58±0.02 | 93.65±0.00  |

## Results on IJB-C
| Method/Data | Aux | {$n^{s}$} | {$n^{r}$} | B-1e-6 | B-1e-5 | B-1e-4 | B-1e-3 | B-0.01 | B-0.1 | Avg |
| ---- |  ---- |  ---- |  ---- |  ---- |  ---- |  ---- |  ---- |  ---- |  ---- |   ---- |
| CASIA-WebFace IR50 | N/A | 0     | $\sim$0.5M | 0.83±0.10 | 5.86±1.31 | 56.87±3.14 | 89.41±0.40 | 96.19±0.06 | 98.61±0.02 | 57.96±0.83  |
| CASIA-WebFace IR101$^{\dagger}$  | N/A | 0     | $\sim$0.5M | 0.38±0.13 | 3.92±1.96 | 55.21±6.21 | 90.42±0.76 | 96.55±0.19 | 98.69±0.10 | 57.53±1.54  |
| Auggen from CASIA IR50 |  $\textcolor{teal}{N}$ | $\sim$0.2M   | $\sim$0.5M  | 1.43±0.22 | 9.67±1.01 | 61.75±3.48 | 90.00±0.44 | 96.17±0.19 | 98.64±0.01 | 59.61±0.81  |
| ---- |  ---- |  ---- |  ---- |  ---- |  ---- |  ---- |  ---- |  ---- |  ---- |   ---- |
| WebFace10K IR50 | N/A | 0 |  $\sim$0.16M  | 70.37±0.75 | 78.81±0.32 | 86.45±0.11 | 92.68±0.01 | 96.52±0.05 | 99.02±0.01 | 87.31±0.20  |
| WebFace10K IR101 $^\dagger$  | N/A | 0 | $\sim$0.16M  | 72.56±0.02 | 81.26±0.14 | 88.27±0.23 | 93.55±0.07 | 97.02±0.07 | 99.12±0.00 | 88.63±0.08  |
| AugGen from WebFace10K IR50 | $\textcolor{teal}{N}$ | 0.2M | $\sim$0.16M | 75.47±0.22 | 82.59±0.01 | 89.35±0.05 | 94.24±0.01 | 97.30±0.02 | 99.18±0.01 | 89.69±0.03  |


We observed a similar improvement for IJB-B but omitted it due to space constraints.


**In conclusion**, these findings show that AugGen is an effective strategy for the evaluated tasks. It should be of interest to the research community. We will add the new results to the paper.
Due to time constraints, we used a smaller generator preset; results could improve with larger networks.

---

### Author Response · Authors · 2024-11-22
**Explanation and Clarification**

# Clarification on the statement of contributions
As requested, we will clarify our contributions to the paper. This paper introduces a sampling method in a generator's condition space, informed by a discriminator trained on the same dataset. This setup is shown to be effective for the **first** time. The augmented dataset (referred to as “AugGen”) empirically outperforms the original dataset, with notable performance gains surpassing architectural improvements (e.g., a discriminator trained with IR50 on the mix of AugGen and the original dataset outperforms IR101 on the original dataset). This underscores the potential of our method in comparison to architectural interventions.


The key contributions include:
* A simple yet novel method and a metric for finding mixing parameters in the **condition space** is demonstrated in **Table 3** and **Algorithm 1**.
* Training with our generated augmentations **outperform architectural improvements**, with **simpler** networks.
* The **first** synthetic FR dataset generation approach uses no additional priors for dataset creation.
* Our method outperforms SOTA methods in multiple benchmarks **using less data**, further exposing the inefficiencies of SOTA methods in IJB-B and IJB-C benchmarks.
* A demonstration at this scale is that a generator can boost discriminator performance using a single real dataset **without** auxiliary methods, **unlike** SOTA approaches.
* We show that existing generative performance metrics (e.g., FD, KD) poorly correlate with FR system outcomes, emphasizing the need for better proxy metrics for generative augmentation.


We believe that these findings about the use of generated data are promising and therefore of significant interest to the research community.



# Further Explanation:


Please refer to this **anonymous** link for illustrations:
https://docs.google.com/document/d/e/2PACX-1vQcpzqHf-YThsUTENRD3ZCzLWGQpePODdLLaCUbdIPX0ayhJO_rbPVhXsuIX7rcVDhFxLxpuGypjr8x/pub

As requested by the reviewers we propose to further explain why our method is working. Here for the baseline, we train a discriminator with the original classes using a margin-based loss. The discriminator learns to distinguish between classes M and N with the cosine distance, $\theta_{before}$.

After mixing the conditions of M and N as per Algorithm 1 (as appeared on the right side of the illustration) and treating the result as a new class, the network is forced to discriminate between this new mixed class (between M and N), leading to a larger cosine distance $\theta_{after}$ between M and N. This enhances the discriminative power, as shown by the improved performance across all 8 benchmarks when training on the mixed augmented and original dataset.

## Verification of Hypothesis (Requested by Reviewer y7jg):

Here, we validate the hypothesis in two ways:
- Calculating the average inter-class Similarity between mixed classes before and after applying AugGen samples, **separation of classes**).
-  Average intra-class similarity and standard deviation, **Compactness of samples within each class**.

### Separation of Classes
We verified this hypothesis experimentally. We compared two models: $F_{\mathrm{AugGen}}$, trained on the Original Dataset + AugGen, and $F_{\mathrm{Before}}$ (baseline), trained only on the Original Dataset. For each AugGen mix (e.g., M-N mix), we calculated the similarity of the mixed classes (M and N) using both models across 10,000 classes, then averaged the absolute similarities. Our hypothesis is confirmed if $F_{\mathrm{AugGen}}$ produces lower average similarity (corresponding to higher $\theta_{\mathrm{ours}}$, as shown in the figure). The results support this:

| AugGen (Ours) | Original (baseline) |
| -----  | -------- |
| 0.06645628 | 0.067214645 |

[Updated, finished averaging over all the dataset]

### Compactness
Like previous section, we used the $F_{\mathrm{AugGen}}$ and $F_{\mathrm{Before}}$ to extract embeddings for Mix classes (like M and N) for all mixed classes ($\sim$ 10K). Here we are reporting the per class average similarity within each class (every two different samples of the same class) and averaged it over all the mix classes. We do the same and reporting `std` for each class and averaged it over all mix classes:

| Metric | AugGen (Ours) | Original (baseline) |
| ------------ | ------------ | -----------|
| Avg `Intra Sim` |  0.54911   | 0.49065  |
| Avg `std`       | 0.12807  | 0.13499 |

Here we are demonstrating that the compactness (please refer to how the ovals are more compact in the illustration ) of the embeddings are increased after applying our AugGen samples, further boosting the effectiveness of the margin losses.

These finding validates the **core idea** driving our work.

---

> ### Author Response · Authors · 2024-12-02
>
> ## Adding More AugGen augmentation makes the model better
> By the above explanation we expect that adding more AugGen classes **enhance** model performance. To verify this, we extended the experiments from **Table 4** (stopping at 10K new samples) to 30K in increments of 10K. The generator for the new dataset was trained over ~230 epochs. The results are summarized below.
>
> ### Results on LFW, CFP-FP, CPLFW, AgeDB, CALFW
> | Method | Aux | $n^{s}$ | $n^{r}$ | LFW | CFP-FP | CPLFW | AgeDB | CALFW | Avg |
> | ---- |  ---- |  ---- |  ---- |  ---- |  ---- |  ---- |  ---- |  ---- |  ---- |
> | WebFace10K IR50                          | $\textcolor{teal}{N}$ | 0          |  $\sim$0.16M | 98.97±0.11     | 87.54±0.06     | 93.40±0.01     | 92.55±0.02     | 90.01±0.04     | 92.50±0.02 |
> | AugGen from WebFace10K IR50 (**10Kx20**) | $\textcolor{teal}{N}$ | $\sim$0.2M |  $\sim$0.16M | 99.22±0.09     | 89.06±0.12     | 95.29±0.14     | 93.55±0.20     | 91.71±0.31     | 93.77±0.04 |
> | AugGen from WebFace10K IR50 (**20Kx20**) |  $\textcolor{teal}{N}$| $\sim$0.4M |  $\sim$0.16M | **99.40±0.02** | 89.17±0.07     | 95.36±0.01     | 93.45±0.02     | 92.42±0.03     | 93.96±0.01 |
> | AugGen from WebFace10K IR50 (**30Kx20**) |  $\textcolor{teal}{N}$| $\sim$0.6M |  $\sim$0.16M | 99.38±0.01     | **89.51±0.04** | **96.02±0.16** | **93.66±0.07** | **92.58±0.08** | **94.23±0.02**  |
>
> Here we are observing that adding more mix classes from our AugGen **enhances** the overall model's performance.
>
> We had the same observations for IJB-B and IJB-C.

---

### Author Response · Authors · 2024-12-03
**Summary of Rebuttal**

Dear All,

In our rebuttal, we:

* Extended experiments to a **new dataset** (as source dataset), confirming consistent improvements across all seven benchmarks [y7jg, 1Ykd].
* Showed that training with our augmentations can **surpass** discriminator **architectural improvements**.
* **Explained** our method’s **underlying dynamics** with experimental validation [y7jg, 1Ykd, 1S4a].
* **Differentiated** our approach from reviewer-referenced methods, emphasizing its novelty [y7jg, 1Ykd, 1S4a, ujs9].

We believe we have addressed the reviewers' concerns comprehensively.

Best regards,

---

### Meta-Review · Area_Chair_R3iS · 2024-12-13

**Metareview:**

The paper proposes to train face recognition models on a mix of real data and generated data from a diffusion model. The paper is well written and performs a comprehensive evaluation. However, the reviewers unanimously agree that the paper is not ready for publication. The main reasons being that the presented improvements are somewhat modest, and the paper still falls short of presenting a rigorous justification for when and how the proposed method works, the incremental novelty compared to a number of related works in this space. The AC agrees with the concerns of the reviewers, and does not see a reason to overturn the unanimous decision of the four reviewers.

**Additional Comments On Reviewer Discussion:**

The reviewers unanimously agree that the paper is not ready for publication yet. Three of the reviewers did not participate in a post-rebuttal discussion. The AC read through the reviews and rebuttals and considers most minor concerns resolved (e.g. evaluation on more datasets), but the main concern remains (also in the opinion of the AC) to be that the presented improvements are somewhat modest, and the paper still falls short of presenting a rigorous justification for when and how the proposed method works.

---

### Decision · Program_Chairs · 2025-01-22

Reject